# Divide-and-Denoise: A Game-Theoretic Method for Fairly Composing Diffusion Models

Abhi Gupta [* 1 2]  Polina Barabanshchikova [* 3 4]  Vikas Garg [4 5]  Samuel Kaski [3 4 6]  Tommi Jaakkola [1 2]

## Abstract

The abundance of pre-trained diffusion models provides an opportunity for composition. Combining several models, however, runs the risk of one model dominating or models disagreeing with each other. Here, we propose Divide-and-Denoise, a method for coordinating multiple pre-trained diffusion models during sampling. Much like managing a specialized workforce, our method creates a fair but efficient division of labor across models. Central to our method is the notion of an allocation which defines the responsibility of each model to every region of the noisy sample. At every timestep, we then denoise by (i) updating the allocation by solving a fair division game, where we divide the sample into regions that maximize total utility under fairness constraints, and (ii) aligning the models with this allocation, where we guide each model to denoise within its assigned region. This leads to a new composite denoising process that evolves in tandem with a division process. We evaluate Divide-and-Denoise on conditional image generation. Across several quality metrics, including the GenEval benchmark, our method outperforms baselines and resolves common failures including missing objects and mismatched attributes. Experiments show that Divide-and-Denoise utilizes each model's expertise without neglecting any other model.

## 1. Introduction

Large-scale diffusion models are changing the game for many disciplines. In robotics, models trained on expert demonstrations can act as long-horizon planners in unseen environments (Xu et al., 2024; Chi et al., 2025; Ajay et al., 2023; Pearce et al., 2023), while in biomedicine, models trained on protein structures can propose candidate therapeutics (Jumper et al., 2021; Corso et al., 2023; Alamdari et al., 2023). These advancements build on a substantial body of work in computer vision, where diffusion models were introduced (Song & Ermon, 2019; Ho et al., 2020a), refined (Geng et al., 2024; Peebles & Xie, 2023; Nichol et al., 2022; Saharia et al., 2022; Burgert et al., 2024), and scaled (Rombach et al., 2022; Sehwag et al., 2025). However, training effective diffusion models demands significant computational resources (Sehwag et al., 2025), carries a large carbon footprint (Sehwag et al., 2025), and often requires task-specific fine-tuning (Fan et al., 2023; Wallace et al., 2024; Chen et al., 2024). Given these challenges, there is a pressing need for effective model reuse, control, and composition (Dhariwal & Nichol, 2021; Ho & Salimans, 2022).

Among the many available models, choosing which one to use is not always obvious. A collection of models may even be used together in order to generate data that no individual model could generate alone. Consider, as a running example, one model trained on images of dogs and another on cats. A common approach is to define a composite distribution as the product or mixture of the 'dog' and 'cat' densities (Liu et al., 2022; Du et al., 2023). Other analytical operations include the harmonic mean and contrast (Garipov et al., 2023), as well as logical operations such as AND (Skreta et al., 2025). Although these operations permit tractable sampling, they are often too simple to preserve the characteristics of each model's distribution when there is conflict. For instance, if models are trained on images of animals appearing mainly in the center, sampling from their product density typically produces incoherent, overlapping dogs and cats.

**Related Work.** Recent work has explored composing text-to-image diffusion models to improve spatial control (Bar-Tal et al., 2023; Du et al., 2023). The typical strategy is to have the user segment an image into spatial regions, assign each region a text prompt (e.g., 'dog' or 'cat'), and then denoise each region with the corresponding model. Although simple to implement, these techniques are restricted to user-defined allocations. This kind of division of labor between models is cumbersome to specify, infeasible to de-

---
*Equal contribution  [1]MIT ORC, USA [2]MIT CSAIL, USA [3]ELLIS Institute Finland [4]Department of Computer Science, Aalto University, Finland [5]YaiYai Ltd [6]Department of Computer Science, University of Manchester, UK. Correspondence to: Abhi Gupta <abhig@mit.edu>.

*Proceedings of the 43rd International Conference on Machine Learning*, Seoul, South Korea. PMLR 306, 2026. Copyright 2026 by the author(s).

fine in many domains (e.g., manually partitioning proteins), and does not take into account the strengths or weaknesses of each model. Bar-Tal et al. (2023) for example assume models can faithfully follow the user-prescribed layout, an assumption that often fails and requires additional forms of guidance (Couairon et al., 2023; Manukyan et al., 2025).

**Contributions.** In order to address these shortcomings, we propose Divide-and-Denoise: a game-theoretic framework for coordinating multiple pre-trained diffusion models. Instead of requiring ground-truth partitions, we infer them online by requiring a division of labor among the models that is both fair and efficient. Our method is fully compositional in the sense that models need not share weights, architectures, or training data, as long as they operate in a latent space of the same dimension. We summarize our contributions below:

- An inference-time algorithm for coordinating multiple pre-trained diffusion models with differing expertise.
- Coupled processes: (i) a *division process* which solves a fair division game for assigning regions to models and (ii) a composite *denoising process* which aligns each model with its assigned region.
- Two formulations of model utilities: a general definition applicable to any conditional diffusion model, and a specific instantiation using attention maps for models with cross-attention conditioning.
- Empirical validation using GenEval (Ghosh et al., 2023), CLIP-Score, VQAScore, and ImageReward metrics showing a team of single-concept models outperform existing composition techniques and multi-concept models.

## 2. Background

### 2.1. Diffusion Samplers

Diffusion models define a forward probability path $q_t$, starting from the data distribution $q_0$, and gradually corrupting data with Gaussian noise. These models are typically provided as time-dependent score networks $s_t(\mathbf{x}; \theta)$ approximating $\nabla_{\mathbf{x}} \log q_t(\mathbf{x})$ at each timestep $t$. To generate new data from diffusion models, we can simulate the denoising process with samplers including DDPM (Ho et al., 2020b), DDIM (Song et al.), and other numerical methods. All these procedures can be expressed as sampling from a sequence of Gaussian transition kernels

$$p_t(\mathbf{x}_{t-1}|\mathbf{x}_t; \theta) := \mathcal{N}(\mu_t(\mathbf{x}_t; \theta), \sigma_t^2 I), \quad t = 1, \dots T - 1.$$

Generation begins by drawing $\mathbf{x}_{T-1} \sim \mathcal{N}(0, I)$. The sampler then iteratively produces $\mathbf{x}_{T-2}, \mathbf{x}_{T-3}, \dots, \mathbf{x}_0$ by applying these transition kernels until a final sample $\mathbf{x}_0$ is obtained. Notably, the family of samplers introduced in (Song et al., 2021) allows both the total number of sampling steps

$T$ and the noise schedule $\sigma_t$ to be varied while keeping the pre-trained model fixed. In this framework, the noise level is parameterized by a scalar $\eta \in [0, 1]$, where $\eta = 1$ recovers the DDPM sampler and $\eta = 0$ yields a fully deterministic trajectory.

### 2.2. Diffusion Conditioning

To generate samples with specific concepts, the score $s_t(\mathbf{x}_t; \theta)$ used during sampling is replaced by a surrogate function $\hat{s}_t(\mathbf{x}_t, \boldsymbol{y}; \theta)$ conditioned on a concept $\boldsymbol{y}$. Classifier guidance (Dhariwal & Nichol, 2021) leverages

$$\nabla_{\mathbf{x}} \log q_t(\mathbf{x}_t|\boldsymbol{y}) = \nabla_{\mathbf{x}} \log q_t(\mathbf{x}_t) + \nabla_{\mathbf{x}} \log q_t(\boldsymbol{y}|\mathbf{x}_t)$$

and defines $\hat{s}_t(\mathbf{x}_t, \boldsymbol{y}; \theta) = s_t(\mathbf{x}_t; \theta) + \omega \nabla_{\mathbf{x}} \log q_t(\boldsymbol{y}|\mathbf{x}_t; \theta)$, where $\log q_t(\boldsymbol{y}|\mathbf{x}_t; \theta)$ is modeled by a pre-trained classifier and $\omega$ is a scaling factor. Alternatively, one can train the diffusion model conditionally so that $s_t(\mathbf{x}_t, \boldsymbol{y}; \theta) \approx \nabla_x \log q_t(\mathbf{x}|\boldsymbol{y})$. Both of these ideas are combined in a technique called classifier-free guidance (Ho & Salimans, 2022). It combines both unconditional and conditional scores to define $\hat{s}_t(\mathbf{x}_t, \boldsymbol{y}; \theta) = s_t(\mathbf{x}_t; \theta) + \omega(s_t(\mathbf{x}_t, \boldsymbol{y}; \theta) - s_t(\mathbf{x}_t; \theta))$.

### 2.3. Image Generation

Diffusion models for conditional image generation are trained on large datasets $\mathcal{D} := \{\mathbf{z}, \boldsymbol{y}\}$, where $\mathbf{z}$ is a high-resolution image and $\boldsymbol{y}$ is a label or a text prompt. In practice, images are encoded into compressed representations $\mathbf{x} = \phi(\mathbf{z}) \in \mathbb{R}^{d \times d \times c}$, and the diffusion model operates in a latent space.

**Attention Maps.** Modern diffusion architectures rely heavily on cross-attention layers to condition generation on text prompts represented by embeddings of length $k$ (Vaswani et al., 2017). The text-to-image model can be expressed as $s_t(\mathbf{x}_t, \boldsymbol{y}; \theta) = f_t(\mathbf{x}_t, \{A_t^{(\ell)}(\mathbf{x}_t, \boldsymbol{y}; \theta)\}; \theta)$, where $\{A_t^{(\ell)} \in \mathbb{R}^{d_\ell \times d_\ell \times k}\}$ is a set of per-layer cross-attention maps. For each spatial location, these maps describe how strongly the model attends to each token in the prompt. Generation can be controlled in creative ways by substituting attention maps $\{A_t^{(\ell)}(\mathbf{x}_t, \boldsymbol{y}; \theta)\}$ with those from another pre-trained model $\{A_t^{(\ell)}(\mathbf{x}_t, \boldsymbol{y}; \theta')\}$ (Hertz et al.). Because layers operate at different resolutions $d_\ell$, the map sizes can vary. Upscaling and averaging these maps across layers creates $k$ saliency maps of size $d \times d$, one for each text token, as shown by Tang et al. (2023).

### 2.4. Fair Division

Dividing $m$ goods among $n$ players is a classical problem in game theory (Amanatidis et al., 2023; Nishimura & Sumita, 2021; Dickerson et al., 2014; Cole et al., 2017; Eisenberg & Gale, 1959; Caragiannis et al., 2019). In the setting of indivisible items, each player $i \in \{1, 2, \dots, n\}$ is allocated

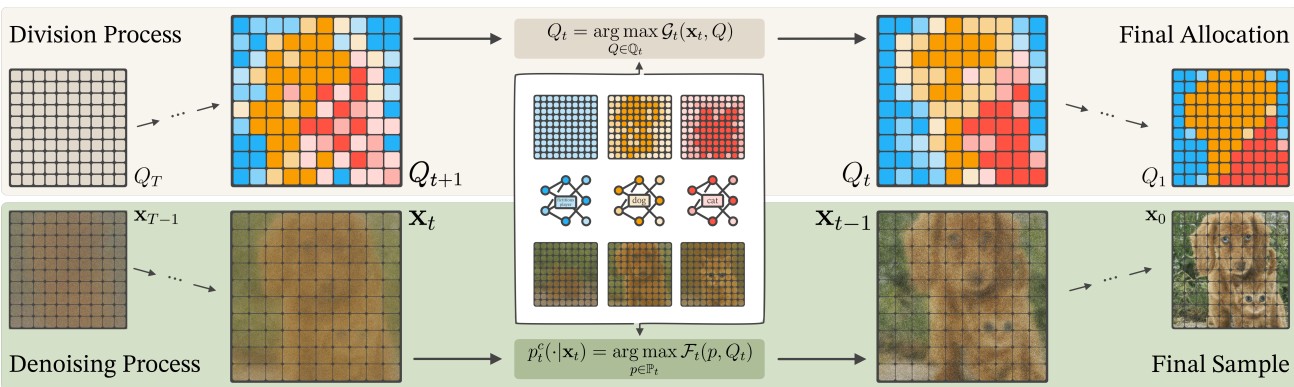

*Figure 1.* Divide-and-Denoise. A noisy image is iteratively refined by two coupled processes: (i) a division step that computes a fair and efficient division of the latent image given by the allocation $Q$ and (ii) a composite denoising step that reconciles the proposals of several single concept diffusion models into a composite update $p$ using this allocation. At every timestep, models provide utilities, the image is divided into soft regions in order to maximize total utility under fairness constraints, and each region is denoised with the assigned model.

a bundle of goods, represented by a binary assignment vector $\mathbf{M}_i \in \{0,1\}^m$, so that no two players share any goods and all goods are allocated. Each player has a utility function $u_i : \{0,1\}^m \to \mathbb{R}_+$ that measures the value of any bundle. Among all possible partitions, we typically seek solutions that are fair and efficient with respect to these utilities. The three main notions of fairness are: *envy-freeness* (no player prefers another player's bundle), *proportionality* (each player receives at least $1/n$ of their total utility), and *equitability* (all players receive bundles of equal utility). Efficiency can be measured, for example, by Nash social welfare, the product of individual utilities.

**Mixed Allocations.** In the case of a single good, no matter who gets it, the partition is not fair to others. This highlights that fair assignments do not always exist. One way to address this challenge is to consider randomized allocations over all possible assignments:

$$\mathbb{M}_{n,m} = \left\{ \mathbf{M} \in \{0,1\}^{n \times m} : \sum_{i=1}^{n} \mathbf{M}_{i,j} = 1 \; \forall 1 \le j \le m \right\}$$

A mixed allocation $Q$ is a discrete distribution over $\mathbb{M}_{n,m}$. Fairness notions are defined in terms of expected utilities under $Q$. For example, an envy-free allocation $Q$ satisfies $\mathbb{E}_{\mathbf{M} \sim Q} u_i(\mathbf{M}_i) \ge \mathbb{E}_{\mathbf{M} \sim Q} u_i(\mathbf{M}_{i'})$ for all $1 \le i \ne i' \le n$. Note that the uniform allocation $\mathcal{U}(\mathbb{M}_{n,m})$ is always fair. Therefore, in the randomized setting, efficiency is crucial to avoid trivial solutions.

**Decomposable Allocations.** When utilities are additive in the goods, $u_i(\mathbf{M}_i) = \sum_{j=1}^{m} u_{ij} \mathbf{M}_{i,j}$, the expected utility of player $i$ simplifies to $\sum_{j=1}^{m} u_{ij} Q_{ij}$, where $Q_{ij} := \mathbb{E}_{\mathbf{M} \sim Q} \mathbf{M}_{i,j}$ is a fractional weight. We say that an allocation $Q$ is *decomposable* if

$$Q(\mathbf{M}) = \prod_{i=1}^{n} \prod_{j=1}^{n} Q_{ij}^{\mathbf{M}_{i,j}} \quad \forall \mathbf{M} \in \mathbb{M}_{n,m}.$$

Decomposable allocations are essentially equivalent to fractional allocations of $m$ divisible goods, where player $i$ receives a fraction $Q_{ij}$ of good $j$.

## 3. Divide-and-Denoise

We study the problem of coordinating $n$ pre-trained diffusion models, each of which operates in a common latent space. We view a latent in this space as a feature map with $m$ features. Each feature need not be a scalar. Without loss of generality, we denote the sequence of denoising kernels for diffusion model $1 \le i \le n$ as follows:

$$p_T^i = \mathcal{N}(0, I), \quad p_t^i(\cdot \mid \mathbf{x}_t) = \mathcal{N}(\mu_t^i(\mathbf{x}_t), \sigma_t^2 I), \; 1 \le t < T.$$

To avoid cluttering of notation, we will often write $p^i(\mathbf{x}_{t-1}|\mathbf{x}_t)$ in place of $p_t^i(\mathbf{x}_{t-1}|\mathbf{x}_t)$.

Additionally, we assume that models have additive preferences over the latent features denoted by $u_{ij}(\mathbf{x}, t)$, i.e. model $i$'s value for feature $j$ of latent $\mathbf{x}$ at timestep $t$. In Section 3.5, we show that all conditional models already possess intrinsic utilities and offer alternatives as well. Our goal is to define a composite denoising process with kernels $p_t^c(\cdot|\mathbf{x}_t)$ that best accounts for the preference of each model.

Models may differ from each other in many ways. In this work, we focus on the case where each model is conditioned on a concept $\mathbf{y}_i$. We expect samples from $p_t^c$ to ideally match what a single model trained on all the concepts appearing together would generate. Such a model, however, may not be available in practice. Note that we do not require the models to share the same architecture or parameters.

### 3.1. Simulating Two Processes

The main components of our approach are outlined in Figure 1. Divide-and-Denoise generates two coupled trajectories: a sampling path of the composite denoising process,

obtained by iteratively drawing $\mathbf{x}_{t-1} \sim p_t^c(\mathbf{x}_{t-1}|\mathbf{x}_t)$, and a path of the division process given by allocations $Q_t$, also obtained by iteratively updating in time. We define each allocation $Q_t$ to be a distribution over $\mathbb{M}_{n,m}$, the space of partitions of $m$ latent features across $n$ models. Since $Q_t$ specifies how the features at time $t$ are distributed among the pre-trained models, it may be interpreted as a division of labor.

We initialize with $Q_T = \mathcal{U}(\mathbb{M}_{n,m})$ and $p_T^c = \mathcal{N}(0, I)$, and draw the first noisy latent as $\mathbf{x}_{T-1} \sim p_T^c$. At each of the remaining timesteps $1 \leq t < T$, we update the allocation and the composite process according to the bi-level optimization:

$$Q_t = \underset{Q \in \mathbb{Q}_t}{\arg\max} \, \mathcal{G}_t(\mathbf{x}_t, Q) \tag{1}$$

$$p_t^c(\cdot|\mathbf{x}_t) = \underset{p \in \mathbb{P}_t}{\arg\max} \, \mathcal{F}_t(p, Q_t) \tag{2}$$

The choice of the optimization objectives $\mathcal{G}$ and $\mathcal{F}$, and the constrained sets $\mathbb{Q}_t$ and $\mathbb{P}_t$ will be discussed in Sections 3.2 and 3.3 respectively. The goal of the first problem (1) is to fairly and efficiently divide the latent among the individual diffusion models, while the purpose of the second problem (2) is to choose a denoising update that best aligns with this division. The optimization objectives are expressed through a common alignment score $U_t$ with a problem-specific regularization. At the end of each timestep, we sample a denoised latent $\mathbf{x}_{t-1}$ from $p_t^c(\cdot|\mathbf{x}_t)$.

### 3.2. Computing a Fair and Efficient Division

We formulate the problem of finding the next allocation (equation 1) as a fair division game, where the goods are latent features and the players are the individual diffusion models. The efficiency of the allocation is measured in terms of the expected total utility

$$U_t(\mathbf{x}, Q) = \mathbb{E}_{\mathbf{M} \sim Q} \sum_{i=1}^{n} \sum_{j=1}^{m} \mathbf{M}_{i,j} u_{ij}(\mathbf{x}, t).$$

The objective $\mathcal{G}_t$ is the efficiency score regularized by a Kullback-Leibler (KL) divergence with positive weight $\beta_t$:

$$\mathcal{G}_t(\mathbf{x}_t, Q) = U_t(\mathbf{x}_t, Q) - \beta_t D_{\mathrm{KL}}(Q\|Q_{t+1}). \tag{3}$$

The regularization term penalizes abrupt changes between consecutive allocations, encouraging temporally smooth allocation trajectories that provide a stable signal for the composite denoising update. A hyperparameter $\beta_t$ controls the trade-off between efficiency and smoothness. For example, when $\beta_t \to \infty$ the allocation $Q_t$ remains uniform throughout generation.

A solution is constrained to lie in the set of fair allocations $\mathbb{Q}_t$. We express this constraint set as

$$\mathbb{Q}_t = \left\{ Q \in \Delta(\mathbb{M}_{n,m}) : \mathbb{E}_{\mathbf{M} \sim Q} \sum_{i=1}^{n} \sum_{j=1}^{m} \mathbf{M}_{i,j} \phi_{ij}(\mathbf{x}_t, t) \preceq \boldsymbol{b} \right\}$$

with coefficients $\phi_{ij}(\mathbf{x}_t, t) = (\phi_{ij}^1(\mathbf{x}_t, t), \ldots, \phi_{ij}^l(\mathbf{x}_t, t))$ and $\boldsymbol{b} = (b_1, \ldots, b_l)$, for all $1 \leq i \leq n$ and $1 \leq j \leq m$, specifying $l$ linear constraints. Despite this simple form, these sets are flexible enough to represent common notions of fairness under additive utilities as shown below.

*Example* 1. Using a single linear inequality $\mathbb{E}_{\mathbf{M} \sim Q} \sum_{i=1}^{n} \sum_{j=1}^{m} \mathbf{M}_{i,j} \phi_{ij}^1(\mathbf{x}_t, t) \preceq b_1$, we can encode the following relations:

1. Setting $b_1 = 0$ and $\phi_{kj}^1(\mathbf{x}_t, t) = -u_{ij}(\mathbf{x}_t, t)I(k = i) + u_{ij}(\mathbf{x}_t, t)I(k = i')$ is equivalent to saying that player $i$ is not envious of player $i'$.

2. Setting $b_1 = 0$ and $\phi_{kj}^1(\mathbf{x}_t, t) = -u_{ij}(\mathbf{x}_t, t)I(k = i) + u_{ij}(\mathbf{x}_t, t)/n$ is equivalent to constraining player $i$ to be allocated at least $1/n$ of its total utility. Alternatively, for the normalized utilities, we can set $b_1 = -1/n$ and $\phi_{kj}^1(\mathbf{x}_t, t) = -u_{ij}(\mathbf{x}_t, t)I(k = i)$.

3. Setting $b_1 = 0$ and $\phi_{kj}^1(\mathbf{x}_t, t) = -u_{ij}(\mathbf{x}_t, t)I(k = i) + u_{i'j}(\mathbf{x}_t, t)I(k = i')$ is equivalent to saying that the allocated utility of player $i$ is greater or equal to that of player $i'$.

Clearly, by stacking inequalities, we can represent envy-free, proportional, and equitable constraints or their combinations for any number of players. It is worth noting that the uniform allocation is always fair, so the feasible set is not empty.

We conclude this section by introducing a generic solution to the optimization problem in equation 1.

**Theorem 3.1.** *Assume that allocation $Q_{t+1}$ is decomposable with weights $Q_{ij}^{t+1}$. Then, the optimal allocation $Q_t$ solving the fair division game (1) is also decomposable with weights*

$$Q_{ij}^t = \frac{\exp(-\langle \lambda^*, \phi_{ij}(\mathbf{x}_t, t) \rangle + u_{ij}(\mathbf{x}_t, t)/\beta_t) Q_{ij}^{t+1}}{Z_j(\lambda^*)}, \tag{4}$$

*where $Z_j(\lambda^*) = \sum_{i=1}^{n} e^{-\langle \lambda^*, \phi_{ij}(\mathbf{x}_t, t) \rangle + u_{ij}(\mathbf{x}_t, t)/\beta_t} Q_{ij}^{t+1}$ is a normalization constant and $\lambda^*$ is a solution of a dual problem that reads*

$$\max_{\lambda \geq 0} \, -\langle \boldsymbol{b}, \lambda \rangle - \sum_{j=1}^{m} \log Z_j(\lambda). \tag{5}$$

### 3.3. Aligning Models with their Allocations

In this section, we address the second problem (equation 2) of selecting a composite denoising kernel $p_t^c$ from the set of

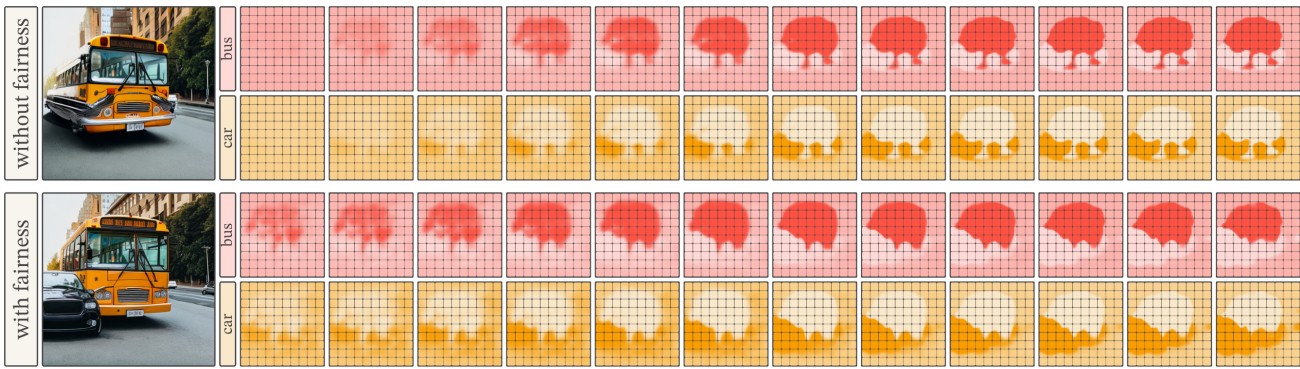

*Figure 2.* The role of fairness in Divide-and-Denoise using Stable Diffusion. Top: without fairness, the car model is allocated significantly fewer pixels than the bus model, resulting in a missing object. Bottom: with fairness, the allocation balances both models so the car is no longer envious of the bus. Fairness prevents a single model from dominating.

feasible denoising distributions $\mathbb{P}_t$. We introduce a new objective that explicitly aligns each diffusion model's proposal with its assigned region, conditioned on the given allocation $Q$. Let $p_j$ denote the marginal of a denoising kernel $p$ corresponding to the feature $j$. We define

$$\mathcal{F}_t(p, Q) = \mathbb{E}_{\mathbf{x}_{t-1} \sim p} U_{t-1}(\mathbf{x}_{t-1}, Q)$$

$$-\alpha_t \mathbb{E}_{\mathbf{M} \sim Q} \left[ \sum_{i=1}^{n} \sum_{j=1}^{m} \mathbf{M}_{i,j} D_{\mathrm{KL}}(p_j(\cdot|\mathbf{x}_t) || p_j^i(\cdot|\mathbf{x}_t)) \right] \quad (6)$$

The aim of the KL regularization is to keep the denoising update for the features allocated to model $i$ close to its proposal. A hyperparameter $\alpha_t > 0$ controls the trade-off between alignment with the given allocation and adherence to the proposals of individual models. Notice how $u_{ij}(\mathbf{x}_{t-1}, t-1)$ contributes to the player's utility $u_i(\mathbf{M}_i)$ only if $\mathbf{M}$ assigns feature $j$ to model $i$. Maximizing $U_{t-1}$ encourages each model to concentrate preference on its allocated region while suppressing preference outside it.

Since the objective in 2 can be non-linear, we cannot find an explicit solution in general. However, a solution exists under simplifying assumptions.

**Theorem 3.2.** *Consider the optimization in equation 2 with the following assumptions: (1) $\mathbb{P}_t$ is a set of all distributions over the latent space that factorize over features and (2) $U_t$ is linear jointly in the first argument and $t$. For each $1 \leq i \leq n$, define the marginal weight vector $Q_i$ as $\mathbb{E}_{\mathbf{M} \sim Q_t} \mathbf{M}_i$.*

*Then, the optimal composite denoising kernel is given by $p_t^c(\cdot|\mathbf{x}_t) = \mathcal{N}(\mu_t^c, \sigma_t^2 I)$, where*

$$\mu_t^c = \sum_{i=1}^{n} \mu_t^i(\mathbf{x}_t) \odot Q_i + \frac{\sigma_t^2}{\alpha_t} \nabla_{\mathbf{x}_t} U_t(\mathbf{x}_t, Q).$$

*Here, $\odot$ denotes feature-wise product.*

Remarkably, the solution naturally decomposes into two parts: a compositional update given by the first term and

a guidance update given by the gradient term. Furthermore, when the influence of the regularization increases $(\alpha_t \to \infty)$, the optimal solution converges to the standard MultiDiffusion (Bar-Tal et al., 2023) update.

In practice, we propose to use a local linearization technique. Applying a first-order Taylor expansion to linearize the reward, we approximate $p_t^c(\cdot|\mathbf{x}_t)$ with $\mathcal{N}(\hat{\mu}_t^c, \sigma_t^2 I)$, where

$$\hat{\mu}_t^c(\cdot|\mathbf{x}_t) = \sum_i \mu_t^i(\mathbf{x}_t) \odot Q_i + \frac{\sigma_t^2}{\alpha_t} \sum_{i,j} Q_{ij} \nabla_{\mathbf{x}_t} u_{ij}(\mathbf{x}_t, t).$$

We observe that performance is sensitive to the hyperparameter $\alpha_t$. Large values of $\alpha_t$ suppress the influence of the guidance term, while overly small values may lead to out-of-distribution samples. We find it useful to reparameterize $\alpha_t$ as

$$\alpha_t = \frac{\sigma_t}{\gamma} \|\nabla_{x_t} U_t(x_t, Q)\|. \quad (7)$$

where $\gamma$ is a constant independent of time. The proposed approach is summarized in Algorithm 1.

### 3.4. Fictitious player

A standard approach to compositional sampling (Liu et al., 2022) combines diffusion processes by computing geometric mean of their denoising kernels, which reduces to averaging the means of the Gaussians. Our experiments demonstrate that this process alone often fails to resolve conflicts between models with overlapping preferences, frequently producing blended concepts. Nevertheless, this behavior can be exploited to promote collaboration between models in low-utility regions. To this end, we augment the set of players with a fictitious process whose denoising kernels are given by

$$p_t^{n+1}(\cdot|\mathbf{x}_t) = \mathcal{N}(\mu_t^{n+1}(\mathbf{x}_t), \sigma_t^2 I),$$

with $\mu_t^{n+1}(\mathbf{x}_t) = \frac{1}{n} \sum_{i=1}^{n} \mu_t^i(\mathbf{x}_t)$. We assign this player a uniform utility, namely $u_{(n+1)j} = 1/m$ for each $j$. Importantly, fairness is enforced only for non-fictitious players.

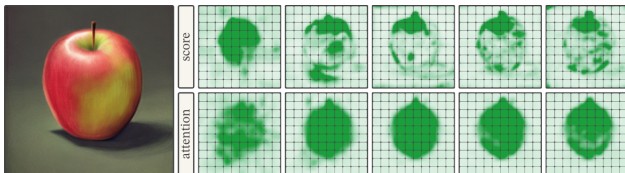

*Figure 3.* Evolution of score-based utilities on top and attention-based on bottom when generating an apple with Stable Diffusion.

### 3.5. Defining Player Utilities

**Score-based Utility**. We view a latent as a feature map $\mathbf{x} \in \mathbb{R}^{m \times c}$ with features $\mathbf{x}^j \in \mathbb{R}^c$ ($c$ channels for images). For $a, b \in \mathbb{R}^{m \times c}$ and a binary vector $\mathbf{M}_i \in \{0, 1\}^m$, let $\langle a, b \rangle_{\mathbf{M}_i} = \sum_{j=1}^m \mathbf{M}_{i,j} (a^j)^\top b^j$ be the inner product restricted to the features selected by $\mathbf{M}_i$. Writing $g_i = \nabla_{\mathbf{x}} q_t(\boldsymbol{y}_i | \mathbf{x}; \theta_i) / \|\nabla_{\mathbf{x}} q_t(\boldsymbol{y}_i | \mathbf{x}; \theta_i)\|$ with blocks $g_i^j \in \mathbb{R}^c$, we define utility of a model $i$ for a bundle $\mathbf{M}_{i'}$ as

$$u_i(\mathbf{M}_{i'}) = \langle g_i, g_i \rangle_{\mathbf{M}_{i'}} = \sum_{j=1}^m \mathbf{M}_{i',j} \, \|g_i^j\|^2,$$

the energy of $g_i$ captured by its assigned features. Appendix A.2 shows that under classifier-free guidance these utilities follow directly from the score:

$$u_{ij}(\mathbf{x}, t) = \frac{\|s_t^j(\mathbf{x}, \boldsymbol{y}_i; \theta_i) - s_t^j(\mathbf{x}; \theta_i)\|^2}{\|s_t(\mathbf{x}, \boldsymbol{y}_i; \theta_i) - s_t(\mathbf{x}; \theta_i)\|^2}.$$

These can be computed for any conditional diffusion model, without building utilities by hand or from data.

**Attention-based Utility**. In the text-to-image setting, cross-attention maps have been shown to be effective indicators of the relevance of each pixel to a target word or phrase. This motivates us to define attention-based utilities as

$$u_{ij}(\mathbf{x}, t) = \frac{A_t^j(\mathbf{x}, \boldsymbol{y}_i; \theta_i)}{\sum_{j=1}^m A_t^j(\mathbf{x}, \boldsymbol{y}_i; \theta_i)},$$

where $A_t^j$ denotes the $j$-th coordinate of the attention map $A_t$ aggregated across layers and text tokens.

Attention-based utilities require the conditioning architecture of the diffusion model to include cross-attention layers. Fortunately, this requirement is met by many models in practice. Figure 3 compares the evolution of score-based and attention-based utilities when generating an image containing a single object. Attention-based utilities exhibit less noise and provide a temporally consistent localization signal.

## 4. Experiments

We test our coordination algorithm on image generation, choosing each diffusion model to be conditioned on a single class or prompt.

---

**Algorithm 1** Divide-and-Denoise

---

**Input:** $n$ pre-trained diffusion models, $\gamma > 0$, $\beta_t > 0$.
Initialize $p_T^c = \mathcal{N}(0, I)$ and $Q_T = \mathcal{U}(\mathcal{M}_{n+1,m})$.
Sample $\mathbf{x}_{T-1} \sim p_T^c$.
**for** $t = T - 1, \ldots, 1$ **do**
    Aggregate $\{p_t^i(\cdot | \mathbf{x}_t) = \mathcal{N}(\mu_t^i(\mathbf{x}_t), \sigma_t^2 I)\}$, $\{u_{ij}(\mathbf{x}_t, t)\}$
    # Division Process
    Solve dual problem in equation 5 for $\lambda^*$
    Update allocation $Q_{ij}^t \propto e^{-\langle \lambda^*, \phi_{ij} \rangle + u_{ij}/\beta_t} Q_{ij}^{t+1}$
    # Composite Denoising Process
    Update mean of the denoising kernel with
    $\hat{\mu}_t^c = \sum_{i=1}^n \mu_t^i(\mathbf{x}_t) \odot Q_i^t + \gamma \sigma_t \frac{\nabla_{\mathbf{x}_t} U_t(\mathbf{x}_t, Q)}{\|\nabla_{\mathbf{x}_t} U_t(\mathbf{x}_t, Q)\|}$
    # Update Sample
    Sample $\mathbf{x}_{t-1} \sim p_t^c(\cdot | \mathbf{x}_t) = \mathcal{N}(\hat{\mu}_t^c(\mathbf{x}_t), \sigma_t^2 I)$
**end for**

---

### 4.1. Models

**Stable Diffusion.** We use Stable Diffusion 2.0 (Rombach et al., 2022), a text-to-image latent diffusion model with cross-attention conditioning and latent dimension $64 \times 64 \times 4$. Here, single concepts are represented by short text descriptions. For a model $i$ conditioned on a single concept $\boldsymbol{y}_i$, we use the prompt "an image with $\boldsymbol{y}_i$". For Stable Diffusion experiments, we use attention-based utilities. Following (Chefer et al., 2023), we aggregate only cross-attention maps at resolution 16. After upscaling, this produces a $64 \times 64$ saliency map for each text token. Out of all saliency maps, we select only the ones corresponding to the tokens associated with $\boldsymbol{y}_i$ and average them to obtain $A_t(\mathbf{x}, \boldsymbol{y}_i; \theta_i)$. For quantitative comparison with score-based utilities, see Appendix A.5.

**DiT.** We also evaluate our method using Diffusion Transformer (DiT) (Peebles & Xie, 2023), a class-conditioned diffusion model trained on ImageNet (Russakovsky et al., 2014). In this setting, each model $i$ is conditioned on a single class $\boldsymbol{y}_i$ from ImageNet, and no multi-concept model is available. For DiT experiments, we use score-based utilities. To reduce noise in the utilities, we apply Gaussian blur with kernel size 5 and clip the utilities prior to normalization.

### 4.2. Coordination Strategies

In all experiments, we employ a DDIM scheduler (Song et al., 2021), setting $T = 50$ sampling steps and a noise scale of $\eta = 0.015$. We use classifier-free guided models with the guidance scale $\omega = 7.5$ for Stable Diffusion and $\omega = 4$ for DiT. Divide-and-Denoise uses hyperparameters $\gamma = \eta$ in equation 7, while in equation 3 we use constant $\beta_t = 0.01$ for score-based utilities and $\beta_t = 0.001$ for attention-based. If not specified otherwise, proportional fairness is applied: each of $n$ models is constrained to receive at least $1/n$ of its total utility. Sensitivity to hyperparameters and choice of

fairness constraints are discussed in Appendices A.4 and A.5 We compare our method against the following baselines:

**Averaging.** We construct a composite denoising process by averaging the scores from each single-concept diffusion model at each generation step. This represents a baseline where the division of labor is uniform.

**Composable Diffusion.** We employ a popular approach to compositional sampling from conditional models (Liu et al., 2022). At each iteration, the scores are aggregated as

$$\hat{s}_t(\mathbf{x}_t; \theta) = s_t(\mathbf{x}_t; \theta) + \sum_{i=1}^{n} \omega_i(s_t(\mathbf{x}_t, \boldsymbol{y}_i; \theta) - s_t(\mathbf{x}_t; \theta)).$$

For this baseline, we set $\omega_i = \omega$ for each $i$. Note that setting $\omega_i = \omega/n$ recovers the averaging baseline.

**Multi-Concept Diffusion.** We construct a composite denoising process with a single, multi-concept diffusion model. Since DiT models cannot simultenously condition on several classes, this baseline applies only to Stable Diffusion where multiple concepts can be combined in a joint text prompt. We avoid enriching the multi-concept prompt with information beyond the concept set $\mathcal{Y} := \{\boldsymbol{y}_1, \ldots, \boldsymbol{y}_n\}$, such as relationships between pairs of concepts, since a team of specialized models would not typically have access to this information. The multi-concept prompt is constructed as "`an image with` $\boldsymbol{y}_1$ `and` $\boldsymbol{y}_2$ `and ... and` $\boldsymbol{y}_n$". In this baseline the division of labor is implied by default through the output of a single Stable Diffusion model.

**MultiDiffusion.** We adapt the compositional sampling technique proposed in (Bar-Tal et al., 2023) to the setting when user-defined allocations are not available. Fixed assignment $\mathbf{M}$ divides the latent into $n$ equal vertical stripes and assigns the $i$-th strip to the $i$-th model. MultiDiffusion then samples from a sequence of transition kernels $p_t^{\text{multi}}(\cdot|\mathbf{x}_t) = \mathcal{N}\left(\sum_{i=1}^{n} \mu_t^i(\mathbf{x}_t) \odot \mathbf{M}_i, \sigma_t^2 I\right)$.

### 4.3. Evaluation Metrics

We evaluate our method along three axes: multi-concept image generation using single-concept models, correct attribute binding for complex concepts, and composition of intentionally conflicting concepts. We quantify first two of these criteria on a popular benchmark for image generation called GenEval (Ghosh et al., 2023). For each concept set $\mathcal{Y}$, we generate several images by changing the random seed. GenEval detects objects from the COCO vocabulary (Lin et al., 2014) and their color, and reports two specific metrics: **%images**: The percentage of images containing all objects given by text prompts; **%prompts**: The percentage of concept sets $\mathcal{Y}$ where at least one generated image contains all objects.

We complement GenEval with three widely-used performance metrics, each defined as $r_k(\mathbf{z}, \boldsymbol{t})$, for an image $\mathbf{z}$ and

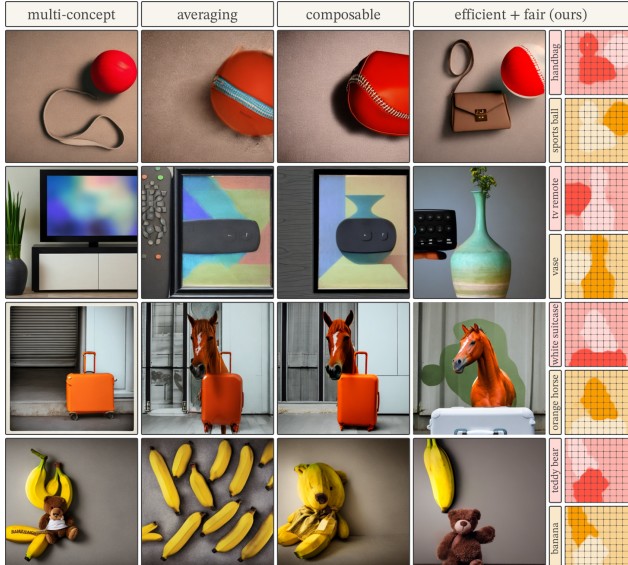

*Figure 4.* Divide-and-Denoise avoids object overlap, preserves all objects, and correctly attributes colors, outperforming baselines on GenEval. Top to bottom: *handbag+sports ball, tv remote+vase, white suitcase+orange horse, teddy bear+banana.*

a prompt $\boldsymbol{t}$. CLIP-Score ($r_1$) measures text–image alignment by computing the cosine similarity between their CLIP embeddings (Radford et al., 2021). Reward ($r_2$) uses a learned reward model trained on human preference data to score how closely the image matches the prompt (Xu et al., 2023). VQA ($r_3$) assesses faithfulness by answering yes-no questions about the prompt and the generated image (Lin et al., 2024). For each metric, we report the following scores averaged across pairs $(\mathcal{Y}, \mathbf{z})$ of a set of concepts $\mathcal{Y}$ and the generated image $\mathbf{z}$: **joint**: $r_k(\mathbf{z}, \boldsymbol{t})$ where $\boldsymbol{t}$ is a multi-concept prompt for $\mathcal{Y}$; **min**: $\min_{\boldsymbol{y}_i \in \mathcal{Y}} r_k(\mathbf{z}, \boldsymbol{t}_i)$ where $\boldsymbol{t}_i$ is a single-concept prompt for $\boldsymbol{y}_i$.

### 4.4. Concepts as Objects

We first evaluate our method along the axis of generating more than one concept. Here each concept is defined as a distinct object. We assess how well Divide-and-Denoise works for multi-concept generation across both Stable Diffusion and DiT setups.

**Stable Diffusion.** For each problem instance, we randomly sample $n$ objects from the COCO vocabulary and define a single-concept prompt for each model $i$ as "`an image with` [$object_i$]". In total, we construct 100 unique $n$-object tuples and evaluate each tuple across 4 different seeds. Results are presented in Table 1, where rows 2 and 3 correspond to $n = 2$ and $n = 3$ object-specific models, respectively. For all metrics except %prompts we report standard error across concept sets. Example images can be found in Figure 4.

We find that improvement over baselines is driven by the

*Table 1.* Performance of Divide-and-Denoise when coordinating 2 and 3 models conditioned on objects using Stable Diffusion.

| Players | Coordination Strategy | GenEval ↑ | | CLIP ↑ | | Reward ↑ | | VQA ↑ | |
|---|---|---|---|---|---|---|---|---|---|
| | | %images | %prompts | joint | min | joint | min | joint | min |
| 2 | Averaging | 31.25%±3.23 | 59% | 26.26±0.27 | 18.64±0.24 | -0.49±0.071 | -1.46±0.048 | 0.720±0.018 | 0.610±0.022 |
| | Composable Diff. | 36.50%±3.26 | 67% | 26.85±0.27 | 18.92±0.27 | -0.26±0.084 | -1.30±0.059 | 0.749±0.021 | 0.643±0.025 |
| | Multi-Concept Diff. | 53.75%±3.32 | 86% | 27.05±0.29 | 18.77±0.25 | 0.28±0.090 | -1.15±0.064 | 0.753±0.022 | 0.683±0.025 |
| | MultiDiffusion | 58.00%±2.95 | 93% | 27.65±0.30 | 19.59±0.22 | 0.34±0.075 | -0.99±0.054 | 0.816±0.016 | 0.738±0.019 |
| | Ours (w/o fairness) | 87.00%±2.15 | 98% | 29.91±0.26 | **21.59**±0.17 | 1.16±0.056 | -0.42±0.043 | 0.959±0.006 | 0.921±0.008 |
| | Ours | **88.50%**±1.86 | **99%** | **30.02**±0.27 | 21.53±0.16 | **1.23**±0.054 | **-0.38**±0.042 | **0.960**±0.007 | **0.925**±0.009 |
| 3 | Averaging | 1.50%±0.69 | 5% | 25.46±0.26 | 16.08±0.16 | -1.32±0.052 | -2.06±0.025 | 0.461±0.017 | 0.296±0.015 |
| | Composable Diff. | 3.50%±0.94 | 13% | 26.82±0.29 | 16.03±0.20 | -1.06±0.064 | -1.97±0.031 | 0.472±0.019 | 0.304±0.017 |
| | Multi-Concept Diff. | 14.75%±2.11 | 43% | 28.45±0.27 | 15.15±0.23 | -0.14±0.086 | -1.82±0.044 | 0.532±0.021 | 0.384±0.021 |
| | MultiDiffusion | 14.00%±2.08 | 37% | 28.05±0.28 | 16.02±0.19 | -0.48±0.078 | -1.79±0.044 | 0.537±0.020 | 0.374±0.021 |
| | Ours (w/o fairness) | 51.75%±3.08 | 88% | 32.68±0.27 | 18.96±0.19 | 1.05±0.066 | -0.92±0.053 | 0.872±0.014 | 0.773±0.018 |
| | Ours | **59.50%**±2.99 | **92%** | **33.21**±0.27 | **19.09**±0.16 | **1.22**±0.056 | **-0.79**±0.048 | **0.921**±0.010 | **0.829**±0.014 |

*Table 2.* Fraction of timesteps with fairness violations when coordinating 2 and 3 models conditioned on objects using Stable Diffusion (mean ± std over 400 trajectories).

| Players | Coordination | Fairness Violation Rate ↓ | |
|---|---|---|---|
| | | envy-free | proportional |
| 2 | MultiDiffusion | 0.485±0.233 | 0.485±0.233 |
| | Ours (w/o fairness) | 0.196±0.118 | 0.306±0.215 |
| | Ours | 0.015±0.008 | 0.039±0.050 |
| 3 | MultiDiffusion | 0.845±0.171 | 0.742±0.217 |
| | Ours (w/o fairness) | 0.356±0.216 | 0.440±0.229 |
| | Ours | 0.017±0.010 | 0.042±0.080 |

efficient division of labor. Fairness improves most metrics further. Observe that the importance of fairness increases as more models participate, since the probability of a model being neglected by an efficient (but not fair) allocation grows. To illustrate the effect of the fairness constraint, we provide a qualitative example in Figure 2. Table 2 reports the fraction of sampling timesteps in which the allocation violates proportional or envy-free fairness when using MultiDiffusion baseline, the unconstrained version of Divide-and-Denoise, and Divide-and-Denoise with proportional fairness constraints. Other baselines are omitted since they do not provide explicit allocation required to evaluate fairness, except for the uniform allocation used by the Averaging baseline, which is fair by definition. The fixed allocation used by MultiDiffusion frequently violates fairness constraints, indicating that some models are consistently dominated by others. Even without explicit fairness constraints, our efficient allocation substantially reduces violations. Enforcing proportional fairness reduces them further, with small residual violations coming from numerical approximations in the dual optimization.

**DiT.** Each concept here is represented as a one-hot encoded ImageNet class depicting an object. We select 15 pairs of

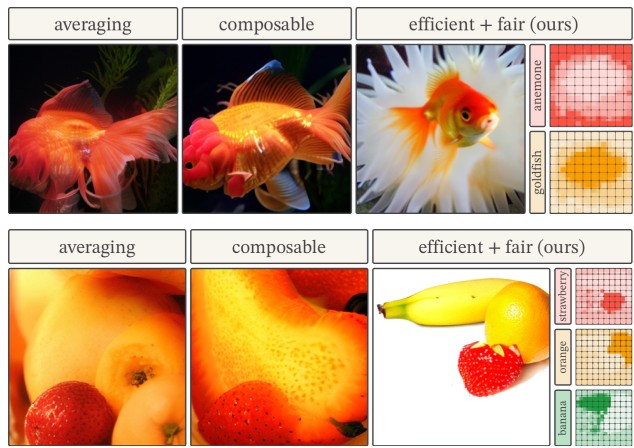

*Figure 5.* Divide-and-Denoise outperforms baselines on ImageNet. Top: *anemone+gold fish*. Bottom: *strawberry+orange+banana*.

*Table 3.* Performance of Divide-and-Denoise using DiT.

| | CLIP ↑ | | Reward ↑ | | VQA ↑ | |
|---|---|---|---|---|---|---|
| | joint | min | joint | min | joint | min |
| Avg. | 25.57 | 19.58 | -1.00 | -1.35 | 0.644 | 0.577 |
| Comp. | 26.67 | 20.43 | -0.69 | -1.11 | 0.700 | 0.634 |
| Multidiff. | 28.56 | 21.73 | 0.23 | **-0.46** | 0.860 | 0.804 |
| Ours | **29.03** | **22.01** | **0.28** | **-0.46** | **0.868** | **0.808** |

objects from the ImageNet-1K dataset (Russakovsky et al., 2014) and evaluate each pair across 20 random seeds. We test our method's ability to coordinate pairs of models to generate images containing both objects. Quantitative results are presented in Table 3, where we compare our method against the Averaging, Composable Diffusion and MultiDiffusion baselines for the $n = 2$ setting.

We note that the multi-concept baseline is unavailable here since no single DiT model can condition on multiple classes. In spite of this, Divide-and-Denoise generates images that

*Table 4.* Performance on coordinating models with descriptive concepts using Stable Diffusion.

| Coordination Strategy | GenEval ↑ | | CLIP ↑ | | Reward ↑ | | VQA ↑ | |
|---|---|---|---|---|---|---|---|---|
| | %images | %prompts | joint | min | joint | min | joint | min |
| Averaging | 9.00%±1.76 | 27% | 28.57±0.26 | 19.81±0.21 | -0.39±0.081 | -1.65±0.051 | 0.641±0.017 | 0.500±0.018 |
| Composable Diffusion | 12.25%±2.03 | 32% | 29.65±0.25 | 20.37±0.23 | -0.09±0.083 | -1.49±0.056 | 0.658±0.019 | 0.522±0.021 |
| Multi-Concept Diffusion | 12.50%±2.37 | 30% | 29.86±0.28 | 19.61±0.26 | 0.10±0.099 | -1.51±0.064 | 0.596±0.021 | 0.455±0.022 |
| MultiDiffusion | 27.00%±3.05 | 58% | 30.05±0.29 | 20.53±0.23 | 0.46±0.092 | -1.19±0.064 | 0.733±0.018 | 0.608±0.022 |
| Ours | **55.75%**±3.44 | **86%** | **32.65**±0.25 | **22.62**±0.17 | **1.34**±0.050 | **-0.56**±0.060 | **0.882**±0.013 | **0.806**±0.017 |

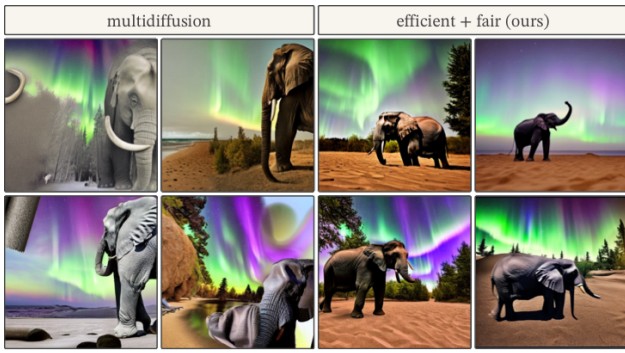

*Figure 6.* Divide-and-Denoise outperforms the fixed allocation of MultiDiffusion when coordinating 3 models with conflicting interests (*sandy beach+aurora borealis+elephant*).

appear as though they were produced by such a multi-class model. We provide qualitative examples for $n = 2$ and $n = 3$ objects in Figure 5.

### 4.5. Concepts with Description

We next evaluate how our method compares to baselines when concepts contain greater detail. This simulates a scenario where a single multi-concept model would typically fail. We attach color descriptions to objects, testing whether our method can faithfully bind attributes to the correct objects. We use Stable Diffusion setup and construct single-concept prompts as earlier, but this time with each concept $y \in \mathcal{Y}$ given by a color and object, e.g. "an *orange horse*". We generate 2 object–color descriptions to define a pair of specialized models. In total, we construct 100 unique pairs and evaluate each pair across 4 different seeds. Results are presented in Table 4. Figure 4 provides a qualitative example of correct attribute binding.

*Table 5.* Performance of Divide-and-Denoise on coordinating models with conflicting interests using Stable Diffusion.

| | CLIP ↑ | | Reward ↑ | | VQA ↑ | |
|---|---|---|---|---|---|---|
| | joint | min | joint | min | joint | min |
| Averaging | 27.81 | 19.14 | -0.36 | -1.45 | 0.687 | 0.521 |
| Composable Diff. | 29.00 | 20.20 | 0.08 | -1.23 | 0.729 | 0.581 |
| Multi-Concept Diff. | 29.76 | 19.82 | 0.65 | -0.88 | 0.752 | 0.633 |
| MultiDiffusion | 29.85 | 20.23 | 0.70 | -0.85 | 0.819 | 0.689 |
| Ours | **31.13** | **21.23** | **1.12** | **-0.55** | **0.905** | **0.815** |

### 4.6. Concepts with Conflict

Finally, we evaluate how well Divide-and-Denoise coordinates models with conflicting interests. To simulate this setting, we hand-design 40 concept sets with naturally competing semantics. For example, we condition the first model on the concept "a desert", while the second on the concept "a snowy mountain". A full list of prompt combinations is provided in Appendix A.6. We use Stable Diffusion setup. As shown in Table 5, Divide-and-Denoise outperforms the coordination baselines. Figure 6 compares images generated by our method against its strongest baseline given by MultiDiffusion with a fixed allocation. By adapting allocations dynamically to model preferences, Divide-and-Denoise produces more diverse spatial layouts and improves overall image quality. Additional examples of generated images can be found in Appendix A.7.

## 5. Conclusion

In this work, we introduced Divide-and-Denoise, a game-theoretic framework for coordinating several pre-trained diffusion models. Our coupled division and denoising processes resolve conflicts between models, prevent concepts or models from being neglected, and outperform baselines across a broad range of metrics including the GenEval benchmark. Notably, our formulation is not tied to a specific model architecture or conditioning mechanism. We achieve compelling results with both DiT and Stable Diffusion, using score-based utilities for the former and attention-based utilities for the latter. Attention-based utilities extend naturally to other domains, including text-to-graph (Chang & Ye, 2025), text-to-audio (Liu et al., 2023), and audio-to-image (Biner et al., 2024) generation. Our score-based utilities can be further applied in domains without cross-attention conditioning. Future work includes designing stronger utility functions, as well as reducing the computational cost of our approach; see Appendix A.3 for discussion. Our results highlight cooperative interaction between pre-trained models as a general recipe for controllable and reusable generative modeling across domains.

## Acknowledgements

AG and TJ acknowledge support from the Machine Learning for Pharmaceutical Discovery and Synthesis (MLPDS) consortium, and the NSF Expeditions grant (award 1918839) Understanding the World through code.

PB and SK were supported by ERC grant ODD-ML 101201120, EU funding ELLIOT 101214398 the Research Council of Finland Flagship programme: Finnish Center for Artificial Intelligence FCAI and decision 341763. SK was supported by the UKRI Turing AI World-Leading Researcher Fellowship (EP/W002973/1).

VG acknowledges Saab-WASP (grant 411025), Research Council of Finland (grant 342077), and the Jane and Aatos Erkko Foundation (grant 7001703) for their support.

## Impact Statement

This paper presents work whose goal is to advance the field of Machine Learning. There are many potential societal consequences of our work, none which we feel must be specifically highlighted here.

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

# A. Appendix

## A.1. Proofs of the Theorems

In our proofs, we will rely on the following technical fact:

**Lemma A.1.** *Let $\pi'$ be a probability distribution over $\mathcal{Z}$, let $\alpha > 0$, and let $f : \mathcal{Z} \to \mathbb{R}$ be a function such that*

$$\int \exp(f(z)/\alpha)\, \pi'(z)dz < \infty.$$

*Then the solution $\pi^*$ to an unconstrained optimization problem*

$$\max_{\pi} \mathbb{E}_{z \sim \pi} f(z) - \alpha D_{\mathrm{KL}}(\pi || \pi')$$

*is given by*

$$\pi^*(z) = \frac{\exp(f(z)/\alpha)\pi'(z)}{\int \exp(f(z)/\alpha)\pi'(z)dz}.$$

*Proof.* It is sufficient to notice that

$$\alpha D_{\mathrm{KL}}(\pi || \pi^*) = -\mathbb{E}_{z \sim \pi} f(z) + \alpha D_{\mathrm{KL}}(\pi || \pi') + C,$$

where $C$ is a constant that does not depend on $\pi$. $\qquad\square$

*Proof of Theorem 3.2.* Recall that denoising kernels of individual models are Gaussians with the same covariance:

$$p^i(\cdot \mid \mathbf{x}_t) = \mathcal{N}(\mu_t^i(\mathbf{x}_t), \sigma_t^2 I).$$

Denote the marginal distribution of $p^i$ corresponding to the latent feature $j$ as $p_j^i(\cdot \mid \mathbf{x}_t) = \mathcal{N}(\mu_j^i(\mathbf{x}_t), \sigma_t^2)$.

For an allocation $Q$ with weights $Q_{ij}$, consider a distribution $p_t^Q$ defined as

$$p_t^Q(\mathbf{x}|\mathbf{x}_t) \propto \prod_{i=1}^{n} \prod_{j=1}^{m} p_j^i(\mathbf{x}^j|\mathbf{x}_t)^{Q_{ij}}.$$

Notice that

$$
\begin{aligned}
p_t^Q(\mathbf{x}|\mathbf{x}_t) &\propto \prod_{i=1}^{n} \prod_{j=1}^{m} p_j^i(\mathbf{x}^j|\mathbf{x}_t)^{Q_{ij}} \\
&= \prod_{i=1}^{n} \prod_{j=1}^{m} \exp\left(\frac{-Q_{ij}\|\mathbf{x}^j - \mu_j^i(\mathbf{x}_t)\|^2}{2\sigma_t^2}\right) \\
&= \exp\left(\frac{-\sum_{i=1}^{n}\sum_{j=1}^{m} Q_{ij}\|\mathbf{x}^j - \mu_j^i(\mathbf{x}_t)\|^2}{2\sigma_t^2}\right) \\
&\propto \exp\left(\frac{-\sum_{i=1}^{n}\sum_{j=1}^{m} [Q_{ij}\|\mathbf{x}^j\|^2 - 2\langle\mathbf{x}^j, \mu_j^i(\mathbf{x}_t)\rangle]}{2\sigma_t^2}\right) \\
&= \exp\left(\frac{-\sum_{j=1}^{m}\|\mathbf{x}^j\|^2 + 2\sum_{j=1}^{m}\langle\mathbf{x}^j, \sum_{i=1}^{n} Q_{ij}\mu_j^i(\mathbf{x}_t)\rangle}{2\sigma_t^2}\right) \\
&= \exp\left(\frac{-\|\mathbf{x}\|^2 + 2\langle\mathbf{x}, \sum_{i=1}^{n} Q_i \odot \mu^i(\mathbf{x}_t)\rangle}{2\sigma_t^2}\right),
\end{aligned}
$$

where $\odot$ denotes feature-wise product, that is $[Q_i \odot \mu^i(\mathbf{x}_t)]_j = Q_{ij}\mu_j^i(\mathbf{x}_t)$. Thus, we obtain

$$p_t^Q(\cdot|\mathbf{x}_t) = \mathcal{N}\left(\mu_t^Q(\mathbf{x}_t), \sigma_t^2 I\right) \quad \text{with} \quad \mu_t^Q(\mathbf{x}_t) = \sum_{i=1}^{n} Q_i \odot \mu_t^i(\mathbf{x}_t).$$

By factorization assumption, for any distribution $p \in \mathbb{P}_t$, it holds that $p(\mathbf{x}) = \prod_{j=1}^{m} p_j(\mathbf{x}^j)$.

Therefore, the following equality holds

$$
\begin{aligned}
\mathbb{E}_{\mathbf{M} \in Q} \sum_{i=1}^{n} \sum_{j=1}^{m} \mathbf{M}_{i,j} D_{\mathrm{KL}}(p_j(\cdot) \| p_j^i(\cdot | \mathbf{x}_t)) &= \sum_{i=1}^{n} \sum_{j=1}^{m} Q_{ij} D_{\mathrm{KL}}(p_j(\cdot) \| p_j^i(\cdot | \mathbf{x}_t)) \\
&= -\sum_{j=1}^{m} \left[ \sum_{i=1}^{n} Q_{ij} \int p_j(\mathbf{x}^j) \log p_j^i(\mathbf{x}^j | \mathbf{x}_t) d\mathbf{x}^j - \sum_{i=1}^{n} Q_{ij} H(p_j) \right] \\
&= -\int p(\mathbf{x}) \log \prod_{i=1}^{n} \prod_{j=1}^{n} p_j^i(\mathbf{x}^j | \mathbf{x}_t)^{Q_{ij}} d\mathbf{x} - \sum_{j=1}^{m} H(p_j) \\
&= -\int p(\mathbf{x}) \log p_t^Q(\mathbf{x} | \mathbf{x}_t) d\mathbf{x} - H(p) + C \\
&= D_{\mathrm{KL}}(p \| p_t^Q(\cdot | \mathbf{x}_t)) + C,
\end{aligned}
$$

where $C$ is a constant that does not depend on $p$.

By Lemma A.1, the composite kernel $p_t^c$ maximizing $\mathcal{F}_t(p, Q)$ is given by

$$
p_t^c(\mathbf{x}_{t-1} | \mathbf{x}_t) \propto \exp(U_{t-1}(\mathbf{x}_{t-1}, Q)/\alpha_t) p_t^Q(\mathbf{x}_{t-1} | \mathbf{x}_t). \tag{8}
$$

Since $U_t(\mathbf{x}, Q)$ is linear jointly in $t$ and $\mathbf{x}$, we have

$$
U_{t-1}(\mathbf{x}_{t-1}, Q) = U_t(\mathbf{x}_t, Q) + A(\mathbf{x}_{t-1} - \mathbf{x}_t) - b,
$$

where $A = \nabla_{\mathbf{x}} U_t(\mathbf{x}_t, Q)$ and $b = \nabla_t U_t(\mathbf{x}_t, Q)$.

Substituting in equation 8, we obtain

$$
\begin{aligned}
p_t^c(\mathbf{x}_{t-1} | \mathbf{x}_t) &\propto \exp(U_{t-1}(\mathbf{x}_{t-1}, Q)/\alpha_t) p_t^Q(\mathbf{x}_{t-1} | \mathbf{x}_t) \\
&= \exp\left( \frac{[U_t(\mathbf{x}_t, Q) + A(\mathbf{x}_{t-1} - \mathbf{x}_t) - b]}{\alpha_t} \right) p_t^Q(\mathbf{x}_{t-1} | \mathbf{x}_t) \\
&\propto \exp\left( \frac{A\mathbf{x}_{t-1}}{\alpha_t} + \frac{-\|\mathbf{x}_{t-1}\|^2 + 2\langle \mathbf{x}_{t-1}, \mu_t^Q(\mathbf{x}_t) \rangle}{2\sigma_t^2} \right) \\
&= \exp\left( \frac{-\|\mathbf{x}_{t-1}\|^2 + 2\langle \mathbf{x}_{t-1}, \mu_t^Q(\mathbf{x}_t) + \sigma_t^2 A/\alpha_t \rangle}{2\sigma_t^2} \right).
\end{aligned}
$$

We conclude that $p_t^c(\cdot | \mathbf{x}_t) = \mathcal{N}(\mu_t^c, \sigma_t^2 I)$ with $\mu_t^c = \sum_{i=1}^{n} \mu_t^i(\mathbf{x}_t) \odot Q_i + \frac{\sigma_t^2}{\alpha_t} \nabla_{\mathbf{x}_t} U_t(\mathbf{x}_t, Q)$.

$\square$

*Proof of Theorem 3.1.* Substituting definition of the efficiency functional $\mathcal{G}$, equation 3, in the fair division game in equation 1, we obtain the following optimization problem

$$
Q_t = \underset{Q \in \mathbb{Q}_t(\mathbf{x}_t)}{\arg\max} \mathbb{E}_{\mathbf{M} \sim Q} \left[ \sum_{i=1}^{n} \sum_{j=1}^{m} \mathbf{M}_{i,j} u_{ij}(\mathbf{x}_t, t) \right] - \beta_t D_{\mathrm{KL}}(Q \| Q_{t+1}), \tag{9}
$$

where $\mathbb{Q}_t = \left\{ Q \in \Delta(\mathbb{M}_{n,m}) : \mathbb{E}_{\mathbf{M} \sim Q} \sum_{i=1}^{n} \sum_{j=1}^{m} \mathbf{M}_{i,j} \phi_{ij} \preceq \boldsymbol{b} \right\}$. Below, we will write $u_{ij}$ meaning $u_{ij}(\mathbf{x}_t, t)$.

By the proof of Lemma A.1, the problem in equation 9 is equivalent to

$$
Q_t = \underset{Q \in \mathbb{Q}_t(\mathbf{x}_t)}{\arg\min} D_{\mathrm{KL}}(Q \| Q^*), \tag{10}
$$

where

$$Q^*(\mathbf{M}) \propto \exp\left(\sum_{i=1}^{n}\sum_{j=1}^{m} \mathbf{M}_{i,j} u_{ij}/\beta\right) Q_{t+1}(\mathbf{M}).$$

$\square$

Assuming that $Q_{t+1}$ is decomposable with weights $Q_{ij}^{t+1}$, we find that

$$Q^*(\mathbf{M}) \propto \prod_{i=1}^{n}\prod_{j=1}^{m} \left(e^{u_{ij}/\beta_t} Q_{ij}^{t+1}\right)^{\mathbf{M}_{i,j}},$$

and thus, the allocation $Q^*(\mathbf{M})$ is decomposable with weights

$$Q_{ij}^* = \frac{e^{u_{ij}/\beta_t} Q_{ij}^{t+1}}{\sum_{i=1}^{n} e^{u_{ij}/\beta_t} Q_{ij}^{t+1}}.$$

We will solve the primal optimization problem in equation 10 in its dual form. Let $\phi(\mathbf{M}) = \sum_{i=1}^{n}\sum_{j=1}^{m} \mathbf{M}_{i,j}\phi_{ij}$. Assuming the set $\mathbb{Q}$ is non-empty (which is always the case for fairness constraints), the corresponding Lagrangian is

$$\max_{\lambda \geq 0, \nu} \min_{Q} \mathcal{L}(Q, \lambda, \nu),$$

where

$$\mathcal{L}(Q, \lambda, \nu) = D_{\mathrm{KL}}(Q||Q^*) + \lambda\left(\mathbb{E}_{\mathbf{M}\sim Q}\phi(\mathbf{M}) - \boldsymbol{b}\right) + \nu\left(\sum_{\mathbf{M}\in\mathbb{M}_{n,m}} Q(\mathbf{M}) - 1\right).$$

Taking derivative with respect to $Q(\mathbf{M})$ we obtain

$$\frac{\partial\mathcal{L}(Q(\mathbf{M}), \lambda, \nu)}{\partial Q(\mathbf{M})} = \log Q(\mathbf{M}) + 1 - \log Q^*(\mathbf{M}) + \lambda\phi(\mathbf{M}) + \nu = 0,$$

and thus,

$$Q(\mathbf{M}) = \frac{Q^*(\mathbf{M})\exp(-\lambda\phi(\mathbf{M}))}{\exp(\nu + 1)}.$$

We can now plug it into the Lagrangian and take derivative with respect to $\nu$:

$$\frac{\partial\mathcal{L}(Q(\mathbf{M}), \lambda, \nu)}{\partial \nu} = \sum_{\mathbf{M}} \frac{Q^*(\mathbf{M})\exp(-\lambda\phi(\mathbf{M}))}{\exp(\nu + 1)} - 1 = 0.$$

Hence, we have

$$Q(\mathbf{M}) = \frac{Q^*(\mathbf{M})\exp(-\lambda\phi(\mathbf{M}))}{Z_\lambda}$$

with $Z_\lambda = \sum_{\mathbf{M}} Q^*(\mathbf{M})\exp(-\lambda\phi(\mathbf{M}))$ and the dual problem reads

$$\max_{\lambda \geq 0} -\log(Z_\lambda) - \langle\boldsymbol{b}, \lambda\rangle.$$

Let $\lambda^*$ be a solution to the dual problem. The optimal allocation is expressed as

$$Q_t(\mathbf{M}) = \frac{Q^*(\mathbf{M})\exp(-\lambda^*\phi(\mathbf{M}))}{Z_{\lambda^*}} \propto \prod_{i=1}^{n}\prod_{j=1}^{m} \left(Q_{ij}^* e^{-\langle\lambda^*, \phi_{ij}\rangle}\right)^{\mathbf{M}_{i,j}}.$$

Hence, it is also decomposable with weights

$$Q_{ij} = \frac{e^{u_{ij}/\beta_t} e^{-\langle\lambda^*, \phi_{ij}\rangle} Q_{ij}^{t+1}}{Z_j(\lambda^*)}$$

where

$$Z_j(\lambda^*) = \sum_{i=1}^{n} \exp\left(-\langle \lambda^*, \phi_{ij} \rangle\right) \exp(u_{ij}/\beta) Q_{ij}^{t+1}.$$

We conclude the proof by noting that $Z_{\lambda^*} = \prod_{j=1}^{m} Z_j(\lambda^*)$.

## A.2. Score-Based Utilities

We derive the simplified form of the score-based utility for feature $j$ under the classifier-free guidance setup. As in the main text, we view a latent as a feature map $\mathbf{x} \in \mathbb{R}^{m \times c}$ with features $\mathbf{x}^j \in \mathbb{R}^c$, and write $s_t^j \in \mathbb{R}^c$ for the corresponding block of the score function. For a vector $v$, let $\bar{v} = v/\|v\|_2$, where $\|\cdot\|_2$ is the Euclidean norm over all features and channels, and $\langle a, b \rangle_{\mathbf{M}_i} = \sum_{j=1}^{m} \mathbf{M}_{i,j} (a^j)^\top b^j$ is the inner product restricted to the features selected by $\mathbf{M}_i$. Writing $g_i = \overline{\nabla_{\mathbf{x}} q_t(\mathbf{y}_i \mid \mathbf{x}; \theta_i)}$ with blocks $g_i^j \in \mathbb{R}^c$, the utility of player $i$ for a bundle $\mathbf{M}_{i'}$ is

$$u_i(\mathbf{M}_{i'}) = \langle g_i, g_i \rangle_{\mathbf{M}_{i'}} = \sum_{j=1}^{m} \mathbf{M}_{i',j} \|g_i^j\|_2^2.$$

This satisfies the additive assumption we posed on utilities, with $u_i(\mathbf{M}_{i'}) = \sum_{j=1}^{m} \mathbf{M}_{i',j} u_{ij}(\mathbf{x}, t)$ and $u_{ij}(\mathbf{x}, t) = \|g_i^j\|_2^2$, the energy of the normalized classifier gradient carried by feature $j$.

In the classifier-free guidance setup, the conditional score is approximated as $\nabla_{\mathbf{x}} \log q_t(\mathbf{x} \mid \mathbf{y}_i; \theta_i) \approx s_t(\mathbf{x}, \mathbf{y}_i; \theta_i)$. The gradient of a density is related to its score by $\nabla_{\mathbf{x}} q_t(\mathbf{x}; \theta_i) = q_t(\mathbf{x}; \theta_i) s_t(\mathbf{x}; \theta_i)$. Using Bayes' rule on the log-densities,

$$\nabla_{\mathbf{x}} \log q_t(\mathbf{y}_i \mid \mathbf{x}; \theta_i) = \nabla_{\mathbf{x}} \log q_t(\mathbf{x} \mid \mathbf{y}_i; \theta_i) - \nabla_{\mathbf{x}} \log q_t(\mathbf{x}; \theta_i),$$

so the gradient of the posterior density is

$$\nabla_{\mathbf{x}} q_t(\mathbf{y}_i \mid \mathbf{x}; \theta_i) = q_t(\mathbf{y}_i \mid \mathbf{x}; \theta_i) \nabla_{\mathbf{x}} \log q_t(\mathbf{y}_i \mid \mathbf{x}; \theta_i) = q_t(\mathbf{y}_i \mid \mathbf{x}; \theta_i) \left[ s_t(\mathbf{x}, \mathbf{y}_i; \theta_i) - s_t(\mathbf{x}; \theta_i) \right].$$

Let $\Delta s_t(\mathbf{x}, \mathbf{y}_i; \theta_i) = s_t(\mathbf{x}, \mathbf{y}_i; \theta_i) - s_t(\mathbf{x}; \theta_i)$ denote the score difference, with $j$-th block $\Delta s_t^j \in \mathbb{R}^c$. Normalizing the gradient, the scalar $q_t(\mathbf{y}_i \mid \mathbf{x}; \theta_i)$ cancels:

$$g_i = \overline{\nabla_{\mathbf{x}} q_t(\mathbf{y}_i \mid \mathbf{x}; \theta_i)} = \frac{q_t(\mathbf{y}_i \mid \mathbf{x}; \theta_i) \Delta s_t(\mathbf{x}, \mathbf{y}_i; \theta_i)}{q_t(\mathbf{y}_i \mid \mathbf{x}; \theta_i) \|\Delta s_t(\mathbf{x}, \mathbf{y}_i; \theta_i)\|_2} = \frac{\Delta s_t(\mathbf{x}, \mathbf{y}_i; \theta_i)}{\|\Delta s_t(\mathbf{x}, \mathbf{y}_i; \theta_i)\|_2}.$$

Taking the $j$-th block, $g_i^j = \Delta s_t^j / \|\Delta s_t\|_2$, so the utility for feature $j$ is

$$u_{ij}(\mathbf{x}, t) = \|g_i^j\|_2^2 = \frac{\|\Delta s_t^j(\mathbf{x}, \mathbf{y}_i; \theta_i)\|_2^2}{\|\Delta s_t(\mathbf{x}, \mathbf{y}_i; \theta_i)\|_2^2} = \frac{\|s_t^j(\mathbf{x}, \mathbf{y}_i; \theta_i) - s_t^j(\mathbf{x}; \theta_i)\|_2^2}{\|s_t(\mathbf{x}, \mathbf{y}_i; \theta_i) - s_t(\mathbf{x}; \theta_i)\|_2^2}.$$

The denominator is the full squared norm over all features and channels, so the utilities are normalized: $\sum_{j=1}^{m} u_{ij}(\mathbf{x}, t) = 1$, since $\sum_{j=1}^{m} \|\Delta s_t^j\|_2^2 = \|\Delta s_t\|_2^2$. This shows that the utility for each coordinate is proportional to the squared difference between the conditional and unconditional scores at that coordinate, normalized by the total squared norm of the score difference across all coordinates. The key insight is that while the gradient of the posterior density includes the posterior probability $q_t(\mathbf{y}_i|\mathbf{x}; \theta_i)$ as a factor, this cancels out when we normalize, leaving only the score difference.

## A.3. Space and Time Analysis

We analyze the computational cost of running Divide-and-Denoise. Our space and time analysis reveals practical techniques for mitigating the overhead of our method relative to baselines. Following the structure of Algorithm 1, each iteration of generation decomposes into three stages: (i) the *Proposal* stage, in which we run a forward pass for each pre-trained diffusion model, and aggregate the per-model denoising kernels $\{p_t^i\}$ and utilities $\{u_{ij}\}$; (ii) the *Division* stage, in which we solve the convex fair-division problem of equation 5 for the optimal allocation $Q_t$; (iii) the *Denoising* stage, in which we form the composite denoising kernel $p_t^c$ and draw the next sample $x_{t-1}$.

**Stage-wise Breakdown.** Table 6 reports the fraction of total wall-clock time and total CPU memory consumed by each stage of running Divide-and-Denoise on a single AWS EC2 G6e instance. We average wall-clock time and memory usage over many runs with each run generating a single sample (i.e. batch size = 1). We compare how space and time scale with the number of coordinated models from $n = 2$ to $n = 4$. We observe two trends. First, the Division stage is the dominant bottleneck as $n$ grows: its share of wall-clock time rises from roughly 12% at $n = 2$ to 57% at $n = 4$, and it accounts for the vast majority of CPU memory across all settings (82% at $n = 2$, growing to 96% at $n = 4$). The high memory footprint of the Division stage reflects the use of an off-the-shelf convex solver that runs on CPU to solve for the optimal allocation. The Proposal and Denoising stages execute entirely on GPU and contribute little to CPU memory usage. Second, the Denoising stage is the most expensive stage at $n = 2$ but its relative share decreases as more models are added. In other words, computing the guidance gradient $\nabla_{x_t} U_t(x_t, Q)$ is more costly than computing the allocation with only two models. The Proposal stage, despite running one diffusion forward pass per model, scales gracefully because each forward pass can be parallelized on the GPU.

*Table 6.* Breakdown of wall-clock time and memory usage when coordinating 2, 3 and 4 models conditioned on objects using Stable Diffusion.

| Players | Fraction of Total Generation Time (%) | | | Fraction of Total CPU Memory (%) | | |
| --- | --- | --- | --- | --- | --- | --- |
| | proposal | division | denoising | proposal | division | denoising |
| 2 | 0.404±0.06 | 0.118±0.13 | 0.462±0.07 | 0.170 ± 0.065 | 0.820 ± 0.069 | 0.013 ± 0.011 |
| 3 | 0.298±0.09 | 0.300±0.23 | 0.382±0.12 | 0.075 ± 0.064 | 0.918 ± 0.070 | 0.007 ± 0.009 |
| 4 | 0.174±0.10 | 0.569±0.25 | 0.244±0.14 | 0.031 ± 0.042 | 0.964 ± 0.047 | 0.004 ±0.006 |

**Runtime Optimization.** The stage-wise breakdown suggests that optimizing the Division stage of Divide-and-Denoise would be most useful. We observe empirically that the allocation $Q_t$ is established early in the denoising trajectory (see Figure 3). We also observe that the guidance gradient minimally changes the image at later steps of generation. This motivates a simple schedule: (i) restrict the Division stage to the first $k_1$ denoising steps, freezing $Q_t$ afterwards and (ii) restrict applying the guidance term in the Denoising stage to the first $k_2$ steps, disabling guidance thereafter. Beyond steps $k_1$ and $k_2$, the composite denoising kernel reduces to the much cheaper averaging update $\mu_t^c = \sum_{i=1}^n \mu_t^i(x_t) \odot Q^i$ with the most recent allocation.

Table 7 reports wall-clock time in milliseconds per iteration of generation under a specific $(k_1, k_2)$ schedule for $T = 50$ DDIM steps. We compare to MultiDiffusion, the strongest naive composition baseline. Setting $k_1 = 15$ approximately halves the Division-stage cost at $n = 3$ and reduces it by roughly $2\times$ at $n = 4$, while setting $k_2 = 15$ similarly reduces the per-step Denoising cost. Applying both of these optimizations, the total per-iteration time of Divide-and-Denoise at $n = 4$ drops from $741.8$ ms to $379.4$ ms – roughly half the unmitigated cost and within $4\times$ of the baseline. Interestingly, the computational overhead of projecting onto the fair allocation set is substantially larger when no guidance is used. This occurs because explicitly steering the models toward a fair and efficient division encourages better separation of interests, often leading to subsequent allocations already being fair. Without guidance, we find that fairness often needs to be imposed at every step during generation.

**Effect on Task Performance.** A natural concern is whether early truncation of the Division and Denoising stages degrades sample quality. Table 8 reports performance on the "Concepts as Objects" setting of Section 4 under matched $(k_1, k_2)$ configurations, keeping all other hyperparameters fixed. Truncating the division schedule to $k_1 = 15$ has only a slight effect across all metrics, consistent with the observation that allocations are primarily determined early in generation. Truncating the guidance schedule to $k_2 = 15$ has a small effect at $n = 2$, but a more pronounced effect at $n = 3$. This reflects the fact that alignment must be enforced explicitly and consistently as the number of models–and therefore the degree of conflict–increases. However, even the most aggressive setting $(k_1, k_2) = (15, 15)$ continues to outperform the strongest baselines (Multi-Concept Diffusion and MultiDiffusion) by large and consistent margins. For example, at $n = 3$, Divide-and-Denoise with $(15, 15)$ achieves $48.5\%$ GenEval %images versus $14.0$–$14.8\%$ for the baselines.

**Further opportunities.** The remaining cost of the Division stage is dominated by calls to the convex solver. Further speedups are possible by using a faster first-order solver, relaxing the convergence tolerance, fixing the number of solver iterations, or warm-starting from $\lambda_{t+1}^*$ to exploit temporal smoothness in the dual variable. We leave a systematic exploration of these solver-level optimizations to future work.

*Table 7.* Breakdown of wall-clock time per iteration of generation process when coordinating 2, 3 and 4 models conditioned on objects using Stable Diffusion.

| Players | Coordination Strategy | Wall-Clock Time Per Iteration (ms) $\downarrow$ | | | |
|---|---|---|---|---|---|
| | | proposal | division | denoising | total |
| | MultiDiffusion | – | – | – | $78.06 \pm 15.44$ |
| | Ours ($k_1 = 50, k_2 = 50$) | $68.18 \pm 0.93$ | $29.36 \pm 67.54$ | $77.88 \pm 0.218$ | $175.42 \pm 67.55$ |
| 2 | Ours ($k_1 = 15, k_2 = 50$) | $68.06 \pm 0.793$ | $16.38 \pm 20.75$ | $77.82 \pm 0.24$ | $162.26 \pm 20.77$ |
| | Ours ($k_1 = 50, k_2 = 15$) | $64.62 \pm 0.883$ | $104.09 \pm 137.21$ | $23.41 \pm 0.198$ | $192.12 \pm 137.21$ |
| | Ours ($k_1 = 15, k_2 = 15$) | $65.12 \pm 0.738$ | $17.36 \pm 21.83$ | $23.72 \pm 0.780$ | $106.20 \pm 21.86$ |
| | MultiDiffusion | – | – | – | $85.03 \pm 21.81$ |
| | Ours ($k_1 = 50, k_2 = 50$) | $79.67 \pm 0.95$ | $131.72 \pm 175.9$ | $101.97 \pm 0.258$ | $313.36 \pm 175.90$ |
| 3 | Ours ($k_1 = 15, k_2 = 50$) | $80.9 \pm 0.851$ | $72.41 \pm 70.11$ | $102.56 \pm 0.278$ | $255.87 \pm 70.12$ |
| | Ours ($k_1 = 50, k_2 = 15$) | $76.05 \pm 0.944$ | $277.25 \pm 270.08$ | $30.88 \pm 0.176$ | $384.18 \pm 270.08$ |
| | Ours ($k_1 = 15, k_2 = 15$) | $79.54 \pm 0.962$ | $71.46 \pm 68.37$ | $30.66 \pm 0.178$ | $181.66 \pm 68.38$ |
| | MultiDiffusion | – | – | – | $92.58 \pm 24.26$ |
| | Ours ($k_1 = 50, k_2 = 50$) | $90.82 \pm 0.922$ | $524.54 \pm 450.32$ | $126.48 \pm 0.263$ | $741.84 \pm 450.32$ |
| 4 | Ours ($k_1 = 15, k_2 = 50$) | $90.6 \pm 1.02$ | $255.21 \pm 143.48$ | $126.56 \pm 0.251$ | $472.37 \pm 143.48$ |
| | Ours ($k_1 = 50, k_2 = 15$) | $86.03 \pm 1.04$ | $746.9 \pm 500.4$ | $38.1 \pm 0.179$ | $871.03 \pm 500.40$ |
| | Ours ($k_1 = 15, k_2 = 15$) | $86.18 \pm 0.816$ | $255.06 \pm 144.752$ | $38.13 \pm 0.171$ | $379.37 \pm 144.75$ |

*Table 8.* Performance of Divide-and-Denoise when coordinating 2 and 3 models conditioned on objects using Stable Diffusion.

| Players | Coordination Strategy | GenEval $\uparrow$ | | CLIP $\uparrow$ | | Reward $\uparrow$ | | VQA $\uparrow$ | |
|---|---|---|---|---|---|---|---|---|---|
| | | %images | %prompts | joint | min | joint | min | joint | min |
| | Multi-Concept Diffusion | 53.75% | 86.00% | 27.05 | 18.77 | 0.28 | -1.15 | 0.753 | 0.683 |
| | MultiDiffusion | 58.00% | 93.00% | 27.65 | 19.59 | 0.34 | -0.99 | 0.816 | 0.738 |
| 2 | Ours ($k_1 = 50, k_2 = 50$) | **88.50%** | **99.00%** | **30.02** | 21.53 | **1.23** | **-0.38** | **0.960** | **0.925** |
| | Ours ($k_1 = 15, k_2 = 50$) | 86.25% | 99.00% | 29.91 | **21.54** | 1.22 | -0.39 | 0.958 | 0.922 |
| | Ours ($k_1 = 50, k_2 = 15$) | 85.25% | 99.00% | 29.79 | 21.38 | 1.18 | -0.42 | 0.953 | 0.910 |
| | Ours ($k_1 = 15, k_2 = 15$) | 83.25% | 99.00% | 29.77 | 21.36 | 1.15 | -0.44 | 0.953 | 0.910 |
| | Multi-Concept Diffusion | 14.75% | 43.00% | 28.45 | 15.15 | -0.14 | -1.82 | 0.532 | 0.384 |
| | MultiDiffusion | 14.00% | 37.00% | 28.05 | 16.02 | -0.48 | -1.79 | 0.537 | 0.374 |
| 3 | Ours ($k_1 = 50, k_2 = 50$) | **59.50%** | **92.00%** | **33.21** | **19.09** | **1.22** | **-0.79** | **0.921** | **0.829** |
| | Ours ($k_1 = 15, k_2 = 50$) | 58.00% | 89.00% | 32.91 | 19.01 | 1.15 | -0.83 | 0.906 | 0.812 |
| | Ours ($k_1 = 50, k_2 = 15$) | 50.00% | 88.00% | 32.53 | 18.60 | 1.07 | -0.90 | 0.864 | 0.761 |
| | Ours ($k_1 = 15, k_2 = 15$) | 48.50% | 86.00% | 32.34 | 18.47 | 0.999 | -0.86 | 0.846 | 0.737 |

## A.4. Hyperparameter Selection

In this section, we analyze the effect of hyperparameters and the sensitivity of our method. Divide-and-Denoise has two main hyperparameters: $\beta_t$, the regularization weight in the division problem, and $\alpha_t$, the regularization weight in the compositional update. We use a constant $\beta_t = \beta$ throughout generation, and we reparameterize $\alpha_t$ via a single scalar $\gamma$ that controls the guidance strength:

$$\alpha_t = \frac{\sigma_t}{\gamma} \left\| \nabla_{x_t} U_t(x_t, Q) \right\|.$$

We find it advantageous to match the guidance magnitude to the noise level, as overly strong guidance can drive intermediate latents off the training manifold. For a DDIM sampler with hyperparameter $\eta$, this corresponds to setting $\eta = \gamma$. Since DDIM typically performs best with $\eta$ near zero, we focus on small values of $\gamma$.

The limiting cases provide intuition for the role of each parameter. As $\beta \to \infty$, the allocation becomes fixed and independent of the utilities. In combination with $\gamma = 0$, this recovers the Averaging baseline. Guidance is not effective when the allocation is nearly uniform, which motivates using $\beta$ in a lower range. In contrast, $\beta \approx 0$ yields an unregularized allocation determined solely by the current utilities. It can cause numerical instability and is sensitive to the noise, which suggests that larger $\beta$ values are preferable for noisy score-based utilities. Setting $\gamma = 0$ disables guidance, while excessively large $\gamma$ leads to off-manifold updates and artifacts.

We evaluate hyperparameter sensitivity on the same experimental setup as in Section 4.4 with $n = 2$ players and Stable Diffusion. Table 9 summarizes the results. In each experiment, we vary either $\gamma$ or $\beta$ while keeping the other parameter fixed to its default value. The default configuration with $\gamma = 0.015$ and $\beta = 0.001$ was hand-picked based on preliminary experiments.

*Table 9.* Performance of Divide-and-Denoise on coordinating 2 models conditioned on different concepts under varying hyperparameters.

| Changed Parameter | GenEval ↑ | | CLIP ↑ | | ImageReward ↑ | | VQA ↑ | |
|---|---|---|---|---|---|---|---|---|
| | %images | %prompts | joint | min | joint | min | joint | min |
| default: $\gamma = 0.015, \beta = 0.001$ | 88.50% | 99% | 30.02 | 21.53 | 1.23 | -0.38 | 0.960 | 0.925 |
| $\gamma = 0$ | 62.25% | 91% | 28.41 | 20.50 | 0.57 | -0.80 | 0.905 | 0.835 |
| $\gamma = 0.01$ | 87.50% | 99% | 29.77 | 21.39 | 1.15 | -0.43 | 0.954 | 0.917 |
| $\gamma = 0.02$ | 88.50% | 99% | 30.02 | 21.57 | 1.25 | -0.37 | 0.963 | 0.931 |
| $\gamma = 0.05$ | 88.25% | 100% | 30.10 | 21.76 | 1.18 | -0.42 | 0.956 | 0.923 |
| $\gamma = 0.1$ | 78.75% | 99% | 29.95 | 21.89 | 0.86 | -0.71 | 0.927 | 0.886 |
| $\beta = 0.0005$ | 89.00% | 99% | 30.01 | 21.50 | 1.24 | -0.39 | 0.960 | 0.928 |
| $\beta = 0.002$ | 88.00% | 99% | 29.96 | 21.57 | 1.23 | -0.38 | 0.965 | 0.929 |
| $\beta = 0.01$ | 84.00% | 97% | 29.82 | 21.51 | 1.31 | -0.49 | 0.960 | 0.921 |

Even without guidance ($\gamma = 0$), Divide-and-Denoise consistently outperforms the baselines, highlighting the benefits of using dynamic allocation. Increasing $\gamma$ initially improves performance, but beyond a certain point performance degrades due to artifacts caused by out-of-manifold updates. Overall, performance is fairly robust to small perturbations around the optimal ranges.

### A.5. Additional Experimental Results

In addition to the experiments in section 4, we perform several supplementary tests. First, we employ our standard GenEval setup with 2 Stable Diffusion models conditioned on different objects, but this time we use score-based utilities instead of attention-based ones. Results are presented in Table 10. We notice that although this choice of utility significantly decreases performance compared to attention-based one, Divide-and-Denoise still outperforms the first three baselines including Multi-Concept Diffusion. However, comparable results to MultiDiffusion suggest that scores may provide insufficient information about concept localization.

*Table 10.* Performance of Divide-and-Denoise on coordinating 2 models conditioned on different concepts with score-based utilities on GenEval.

| Coordination Strategy | GenEval ↑ | | CLIP ↑ | | ImageReward ↑ | | VQA ↑ | |
|---|---|---|---|---|---|---|---|---|
| | %images | %prompts | joint | min | joint | min | joint | min |
| Averaging | 31.25% | 59.00% | 26.26 | 18.64 | -0.49 | -1.46 | 0.720 | 0.610 |
| Composable Diffusion | 36.50% | 67.00% | 26.85 | 18.92 | -0.26 | -1.30 | 0.749 | 0.643 |
| Multi-Concept Diffusion | 53.75% | 86.00% | 27.05 | 18.77 | 0.28 | -1.15 | 0.753 | 0.683 |
| MultiDiffusion | 58.00% | **93.00%** | 27.65 | 19.59 | **0.34** | **-0.99** | 0.816 | 0.738 |
| Ours (without fairness) | 45.00% | 85.00% | 27.29 | 19.82 | -0.08 | -1.23 | 0.808 | 0.715 |
| Ours (with proportional fairness) | **59.00%** | 91.00% | **28.13** | **20.38** | 0.26 | -1.01 | **0.863** | **0.783** |

Moreover, we analyze how the performance of Divide-and-Denoise is affected by the choice of the fairness constraints. Across all tasks for the Stable Diffusion setup, we compare our method without any regularization (Efficient Allocation), with only proportional constraints (Efficient + Proportional Allocation), with proportional and equitable constraints (Efficient + Proportional + Equitable Allocation), and finally with proportional and envy-free constraints (Efficient + Proportional + Envy-Free Allocation). Note that we only use last option when coordinating 3 models, since in 2 players' case any proportional allocation is also envy-free. We report all metrics in Table 11.

*Table 11.* Performance of Divide-and-Denoise in coordinating 2 and 3 models on different concepts under varying fairness criteria.

| Task | Allocation | GenEval ↑ | | CLIP ↑ | | Reward ↑ | | VQA ↑ | |
|------|-----------|-----------|-----------|--------|--------|----------|--------|--------|--------|
| | | %images | %prompts | joint | min | joint | min | joint | min |
| 2 objects | Efficient | 87.00% | 98.00% | 29.91 | 21.59 | 1.16 | -0.42 | 0.959 | 0.921 |
| | Efficient + Proportional | **88.50%** | **99.00%** | 30.02 | 21.53 | **1.23** | **-0.38** | **0.960** | **0.925** |
| | Efficient + Proportional + Equitable | 87.00% | 98.00% | **30.08** | **21.67** | 1.20 | -0.41 | **0.960** | 0.921 |
| 3 objects | Efficient | 51.75% | 88.00% | 32.68 | 18.96 | 1.05 | -0.92 | 0.872 | 0.773 |
| | Efficient + Proportional | 59.50% | 92.00% | 33.21 | 19.09 | 1.22 | -0.79 | **0.921** | 0.829 |
| | Efficient + Proportional + Equitable | **63.00%** | **94.00%** | **33.24** | **19.35** | **1.29** | **-0.74** | 0.919 | **0.834** |
| | Efficient + Proportional + Envy-Free | 60.25% | 92.00% | 33.17 | 19.05 | 1.24 | -0.77 | 0.915 | 0.824 |
| 2 objects with color | Efficient | 52.50% | **86.00%** | 32.42 | 22.53 | 1.22 | -0.66 | 0.869 | 0.791 |
| | Efficient + Proportional | **55.75%** | **86.00%** | **32.65** | **22.62** | **1.34** | **-0.56** | **0.882** | **0.806** |
| | Efficient + Proportional + Equitable | 54.00% | 85.00% | **32.65** | **22.62** | 1.31 | **-0.56** | **0.882** | 0.804 |
| concepts with conflict | Efficient | - | - | 30.99 | 21.01 | 1.05 | -0.60 | 0.896 | 0.796 |
| | Efficient + Proportional | - | - | 31.13 | 21.23 | 1.12 | -0.55 | **0.905** | 0.815 |
| | Efficient + Proportional + Equitable | - | - | **31.36** | **21.46** | **1.15** | **-0.52** | 0.904 | **0.819** |
| | Efficient + Proportional + Envy-Free | - | - | 31.19 | 21.26 | 1.12 | -0.54 | **0.905** | **0.819** |

## A.6. Custom Dataset

We constructed a custom dataset comprising 40 examples, each designed with conflicts between individual prompts. Specifically, the first 30 prompts involve either object + semantics or semantics + semantics compositions: 10 focus on conflicting attributes (e.g., "an image with a blue lake" and "an image with violet trees"), while the remaining 20 capture conflicting semantic combinations (e.g., "an image with a desert" and "an image with a snowy mountain"). The last 10 prompts are of the form semantics + semantics + object, where at least one pairing is conflicting.

**Conflicting Attributes (10 prompts):**

1. `an image with a blue lake,` `an image with violet trees`

2. `an image with a green car,` `an image with a pink forest`

3. `an image with a yellow elephant,` `an image with a grey desert`

4. `an image with a rainbow-colored dog,` `an image with a black-and-white city`

5. `an image with a brown flamingo,` `an image with a purple swamp`

6. `an image with a transparent car,` `an image with a glowing forest`

7. `an image with a golden cloud,` `an image with a black ocean`

8. `an image with a white bus,` `an image with a bright orange snowfield`

9. `an image with a blue cat,` `an image with a black sofa`

10. `an image with a red eagle,` `an image with a grey sky`

**Conflicting Semantic Combinations (20 prompts):**

11. `an image with a desert,` `an image with a snowy mountain`

12. `an image with a jungle,` `an image with an icy glacier`

13. `an image with a burning forest,` `an image with a frozen river`

14. an image with a tropical beach, an image with a volcanic eruption

15. an image with a futuristic city, an image with a medieval castle

16. an image with a stormy sky, an image with a calm lake

17. an image with a carnival, an image with a haunted graveyard

18. an image with an underwater city, an image with a floating island

19. an image with a sunny meadow, an image with a meteor shower

20. an image with a winter tundra, an image with a blooming spring forest

21. an image with snowy mountains, an image with a camel

22. an image with a rainforest, an image with a penguin

23. an image with a rocky cliffside, an image with a telephone booth

24. an image with an iceberg, an image with a windmill

25. an image with a busy highway, an image with a deer

26. an image with a street, an image with a jaguar

27. an image with a blizzard, an image with a giraffe

28. an image with a desert oasis, an image with a moose

29. an image with a rice field, an image with a Ferris wheel

30. an image with a savanna, an image with a skyscraper

**Triple Combinations (10 prompts):**

31. an image with a desert canyon,    an image with snowy peaks,    an image with a tropical parrot

32. an image with a modern city skyline, an image with a rural farm, an image with a horse

33. an image with a sunflower field, an image with snowy mountains, an image with a polar bear

34. an image with a grassy soccer field, an image with volcanic ash clouds, an image with a motorcycle

35. an image with a frozen lake,   an image with a tropical beach,   an image with a palm tree

36. an image with an iceberg, an image with stormy skies, an image with a cow

37. an image with a grassy meadow,   an image with a tropical sun,   an image with a snowman

38. an image with a sandy beach, an image with the aurora borealis, an image with an elephant

39. an image with a wheat field,   an image with a futuristic glass dome city,   an image with a steam train

40. an image with a rocky cliffside,   an image with a rainbow sky,   an image with a boat

### A.7. Additional Qualitative Results

We provide additional qualitative comparisons of Divide-and-Denoise with baselines. The experimental setups are described in Section 4.

For the experiments using Stable Diffusion, the images are shown in Figures 8, 9, 10, and 11. Each row corresponds to one coordination mechanism: Averaging, Composable Diffusion, Multi-Concept Diffusion, MultiDiffusion and Divide-and-Denoise with fairness given by proportional constraints. Each column corresponds to a fixed set of concepts. For each combination of method and concept set, we generate a batch of 4 images.

For the DiT setup, the images are shown in Figure 7. We present all pairs of ImageNet classes used for the quantitative analysis (see Figure 3) and plot images generated by Composable Diffusion and MultiDiffusion baselines alongside images generated by our method for the same random seed. Divide-and-Denoise avoids object overlap and blending of concepts in many cases where Composable Diffusion fails to reliably represent both classes.

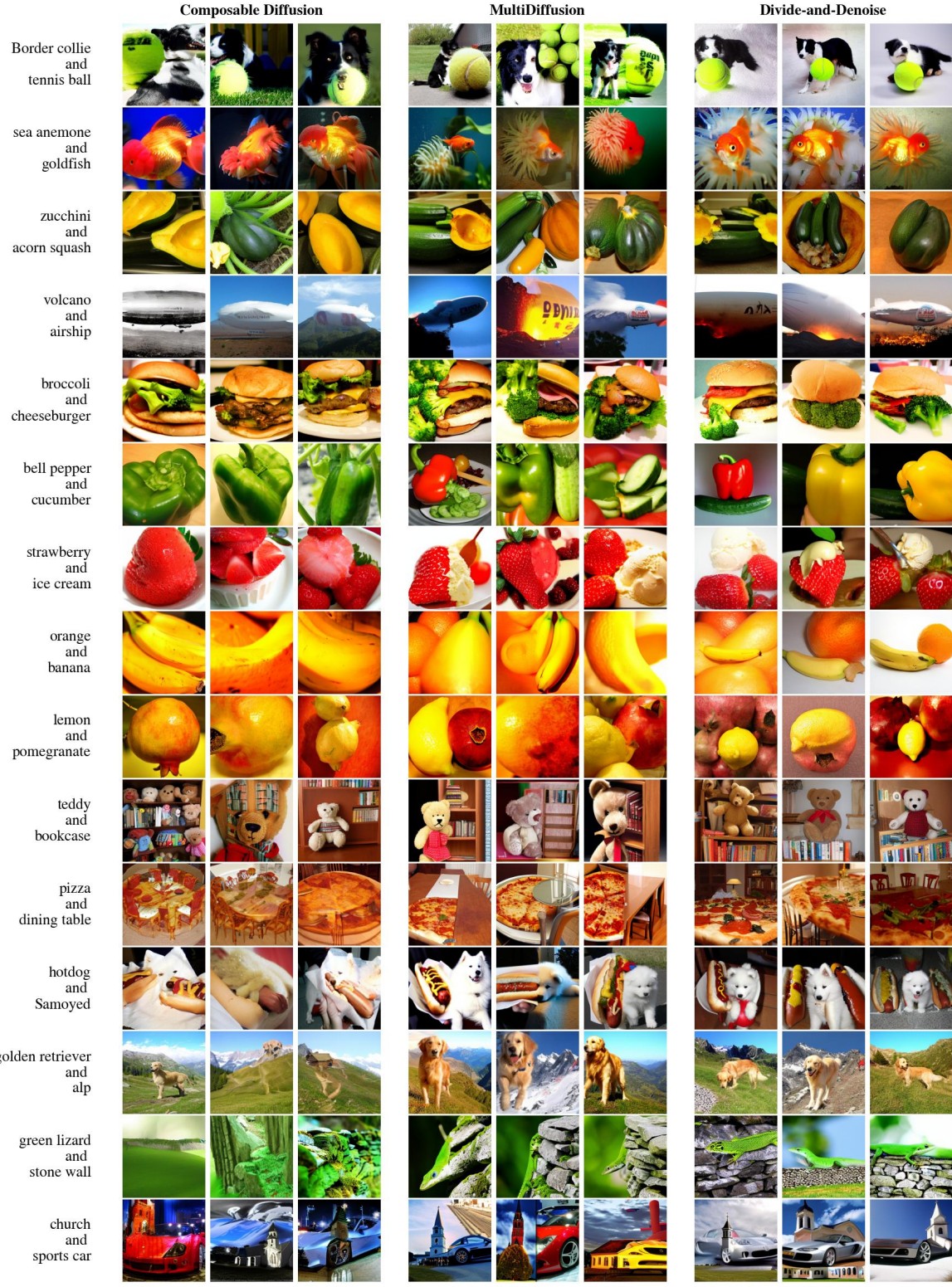

*Figure 7.* Comparison of Composable Diffusion (3 first columns), MultiDiffusion (3 middle columns) and Divide-and-Denoise (3 last columns) on a dataset of ImageNet pairs. For each pair of concepts, we show 3 images generated with random seeds. All three models use the same seeds.

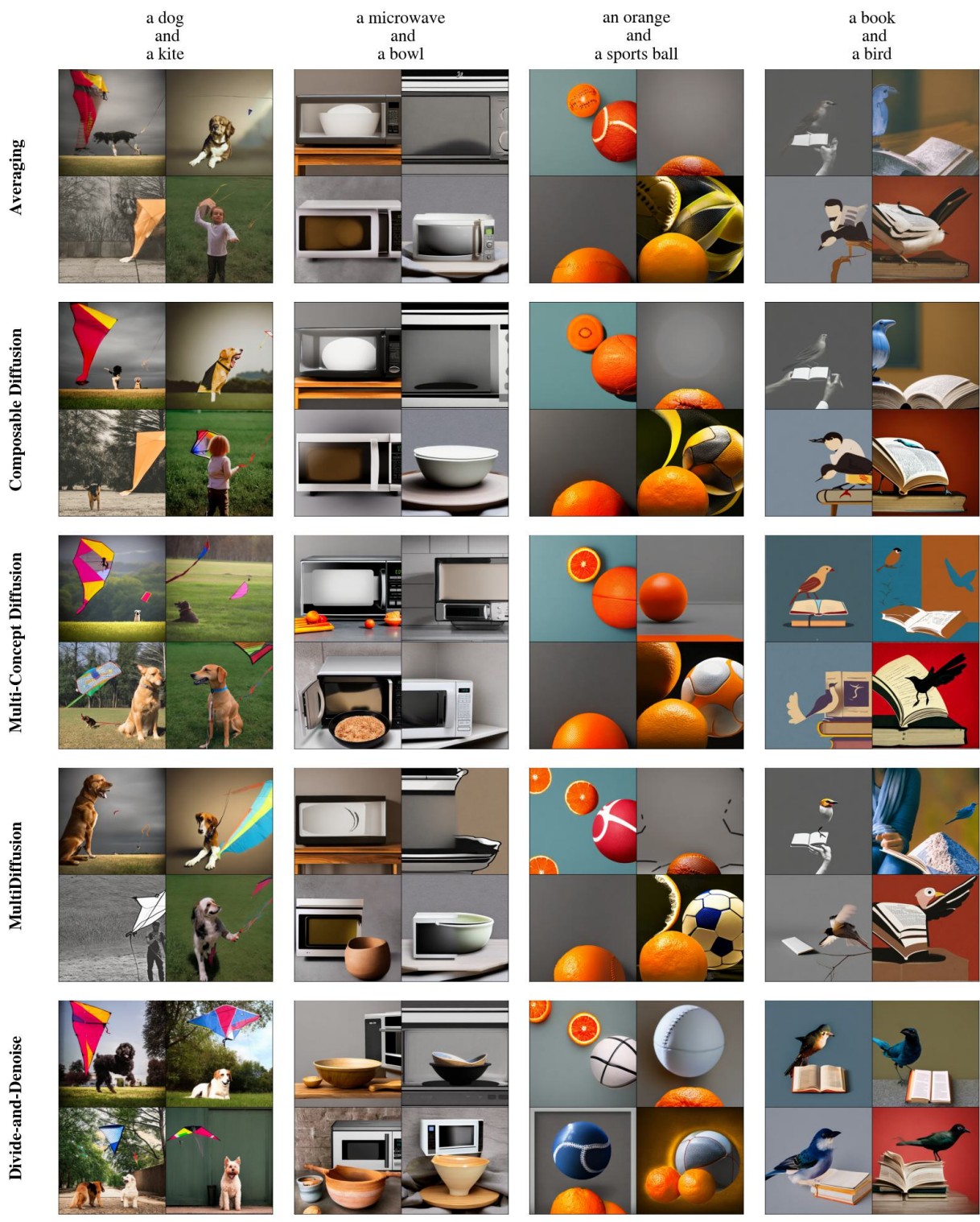

*Figure 8.* Qualitative comparison on GenEval benchmark (2 objects)

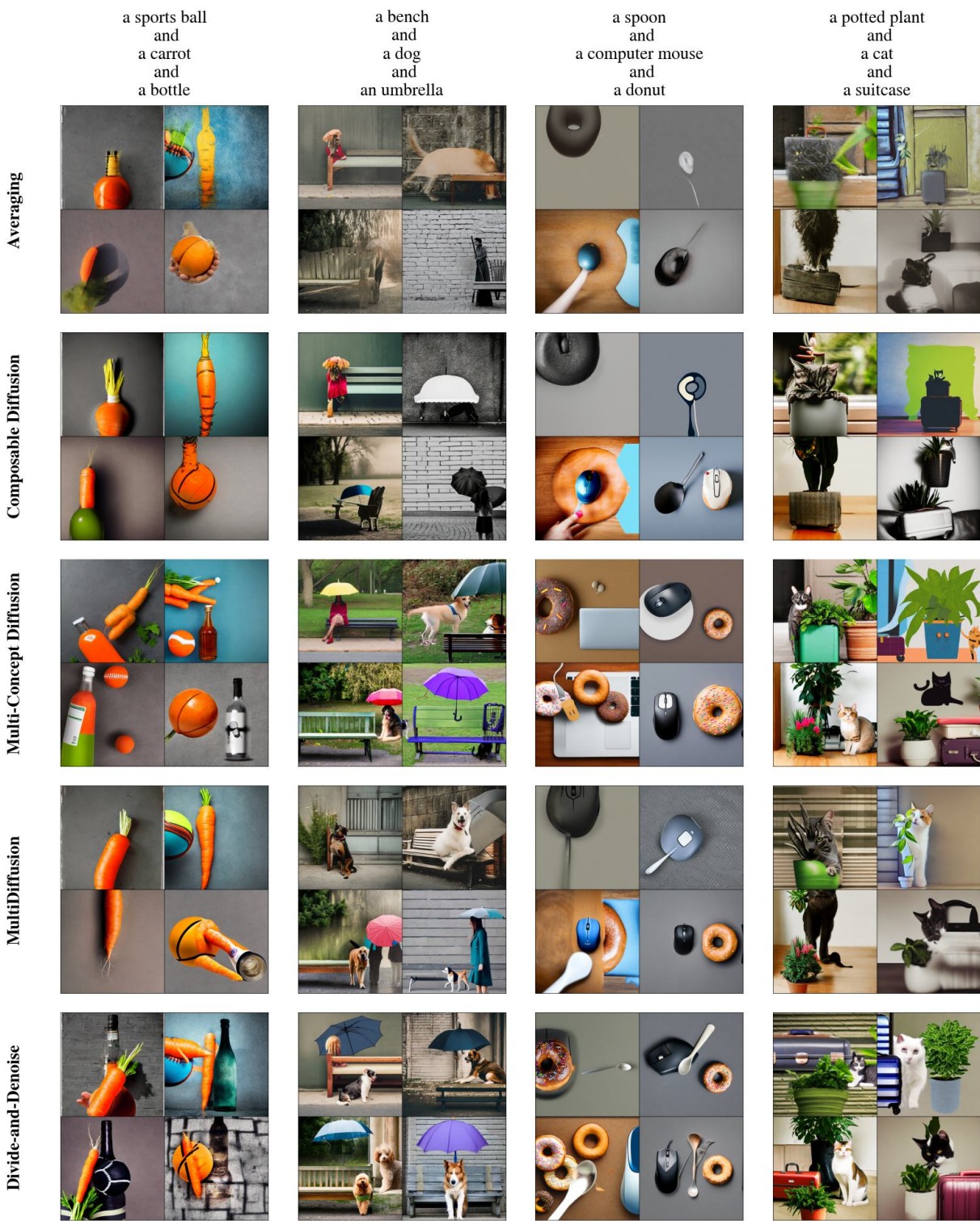

*Figure 9.* Qualitative comparison on GenEval benchmark (3 objects)

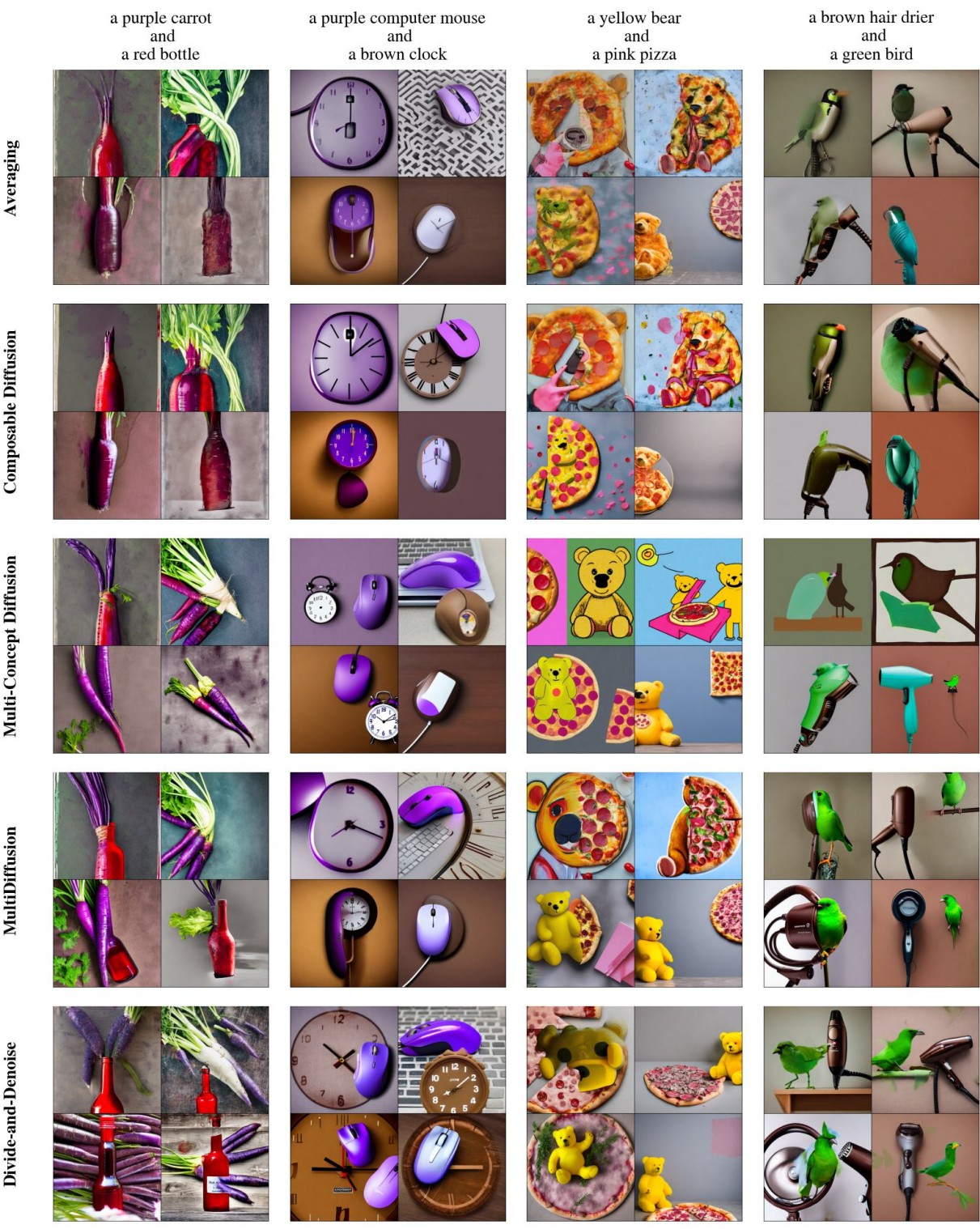

*Figure 10.* Qualitative comparison on GenEval benchmark (2 objects with color descriptions)

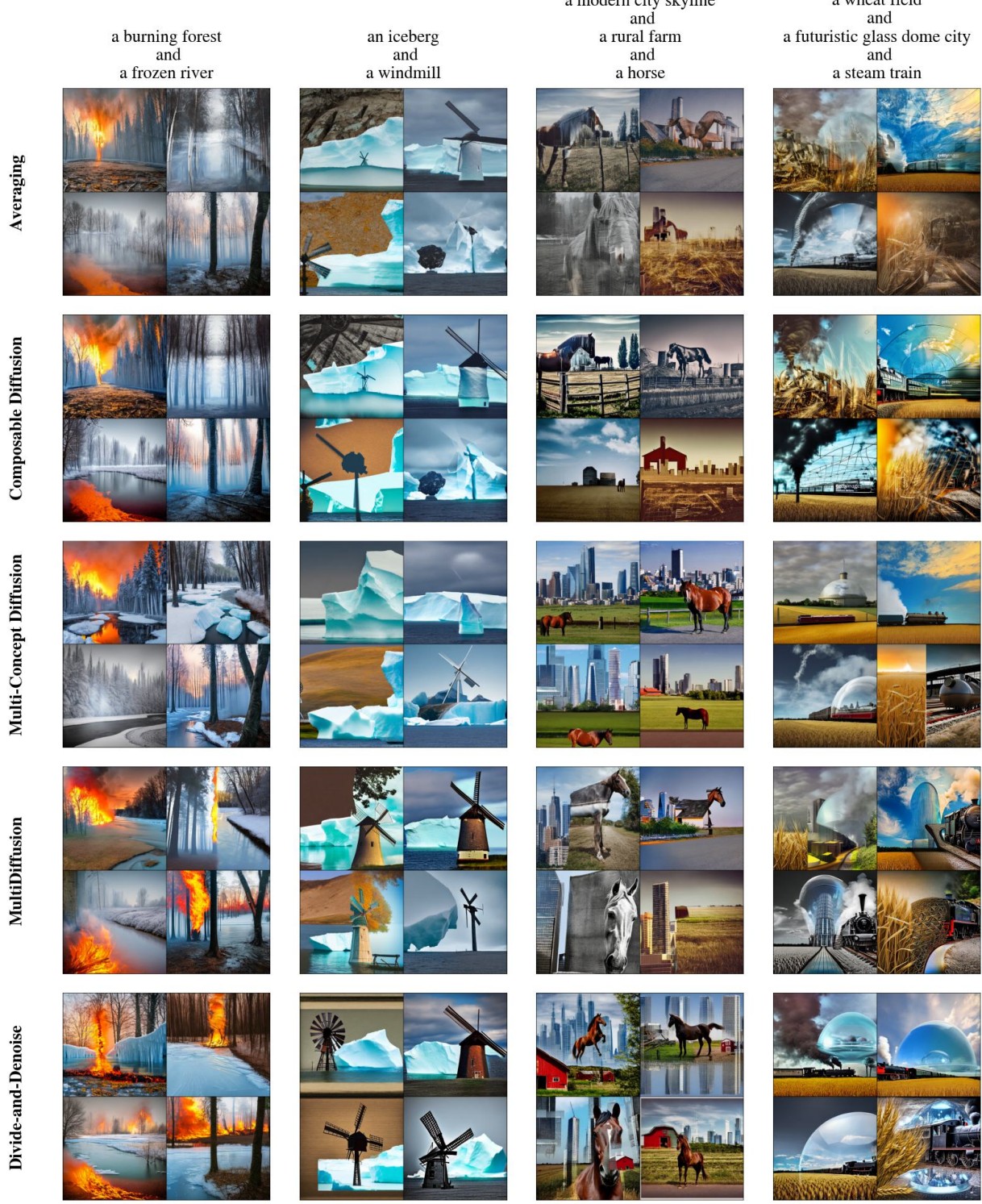

*Figure 11.* Qualitative comparison on Conflict dataset

