# OpenReview forum: "Divide-and-Denoise: A Game-Theoretic Method for Fairly Composing Diffusion Models"
_ICML.cc/2026/Conference — ICML 2026 spotlight_

### Official Review · Reviewer_sVXN · 2026-03-08

**Soundness:** 3
**Presentation:** 4
**Significance:** 3
**Originality:** 3
**Overall Recommendation:** 5
**Confidence:** 3

**Summary:**

The paper discusses the problem of composing multiple diffusion models to synthesize a single image. Since each model can be vastly different (data, architecture), averaging predictions is not enough to produce realistic results. The authors propose a game-theoretic approach where models compete to denoise parts of the image. By solving an allocation problem and composing predictions based on these allocations, both the overall image quality and the adherence of the final image to each model's conditions improve.

**Compliance With Llm Reviewing Policy:**

Affirmed.

**Final Justification:**

The rebuttal addressed my concerns. I am raising my score to reflect this.

**Key Questions For Authors:**

- How costly is solving the allocation problem at every iteration?

- Do allocations change significantly over all timesteps? Could you get away with not solving this problem at every step?

**Limitations:**

The authors have not discussed the limitations of the proposed algorithm.

**Strengths And Weaknesses:**

### Strengths
- The proposed game-theoretic view of composing multiple diffusion models has not been discussed in the literature before. It constitutes an interesting alternative to previous works that treated model composition through the lens of probability theory/energy functions. Additionally, the authors also provide an interesting connection to MultiDiffusion, which has been extensively used to compose multiple model score predictions into a single generated image. Overall, the theoretical contribution of the paper is significant and could lead to future works that dive deeper into this game-theoretic approach.

- The experiments clearly indicate that the proposed method performs better than the naive (or weighted) averaging and composing concepts through text conditioning. The comparisons to the baselines are extensive, spanning multiple settings (text conditioning, class conditioning) and evaluation metrics.

- The presentation of the paper is clear and it can be easily followed by readers not familiar with game theory concepts.

### Weaknesses

- The main weakness of the paper is the lack of a baseline that has the ability to compose the predicted scores in a spatially varying manner. The proposed algorithm solves the allocation problem to figure out "where" in the image each model should apply. All baselines shown in the experiments treat the entire map of predicted scores equally, i.e., without an ability to choose where each model will apply. The Multi-concept diffusion may be able to do it, but it relies solely on the internal cross-attention mechanisms, which is unfair since the discussion focuses on an external composition of different models.

  Since the authors have drawn a nice parallel to MultiDiffusion, which does exactly the thing that the baselines don't, I believe that it should be included as a baseline in the experiments. There are multiple interesting questions to answer by having a stronger baseline that can compete with the proposed algorithm.

  Solving the allocation problem at every timestep, the proposed method figures out the spatial relationship between the different concepts during generation. If that is indeed true, then a baseline using MultiDiffusion with pre-computed locations for each concept should lead to less realistic images. For example, if you always place the objects on the left and right sides of the image, then when trying to compose a dog and a kite (as in your experiments), MultiDiffusion should have a harder time since kites are usually floating above the dogs.

  Overall, it seems that solving the allocation dynamically during generation is the strong point of the proposed algorithm. However, the experiments performed only moderately highlight this advantage. The paper could benefit from a stronger baseline in the experiments.

---

> ### Author Rebuttal · Authors · 2026-03-29
>
> **Q1: MultiDiffusion Baseline**
> We now include a MultiDiffusion-style baseline that divides the image into  $n$ equal vertical strips, with the $i$-th strip denoised by the $i$-th model. The tables below compare this baseline to our method on 2- and 3-object composition tasks for Stable Diffusion and DiT. Divide-and-Denoise outperforms MultiDiffusion across all metrics, with the largest gap at 3 players.
>
> | Players | Method | GenEval %img | GenEval %prt | CLIP jnt | CLIP min | Reward jnt | Reward min | VQA jnt | VQA min |
> | :---- | :---- | :---- | :---- | :---- | :---- | :---- | :---- | :---- | :---- |
> | 2 | Multidiffusion | 58.00% | 93.00% | 27.65 | 19.59 | 0.34 | \-0.99 | 0.816 | 0.738 |
> | 2 | Ours | **88.50%** | **99.00%** | **30.02** | **21.53** | **1.23** | **\-0.38** | **0.960** | **0.925** |
> | 3 | Multidiffusion | 14.00% | 37.00% | 28.04 | 16.02 | \-0.477 | \-1.789 | 0.536 | 0.374 |
> | 3 | Ours | **59.50%** | **92.00%** | **33.21** | **19.09** | **1.22** | **\-0.79** | **0.921** | **0.829** |
>
> | Method | CLIP jnt | CLIP min | Reward jnt | Reward min | VQA jnt | VQA min |
> | :---- | :---- | :---- | :---- | :---- | :---- | :---- |
> | Composable (best prior baseline) | 26.67 | 20.43 | \-0.69 | \-1.11 | 0.700 | 0.634 |
> | Multidiff. | 28.25 | 21.46 | 0.120 | \-0.53 | 0.856 | 0.789 |
> | Ours | **29.03** | **22.01** | **0.28** | **\-0.46** | **0.868** | **0.808** |
>
> MultiDiffusion with a non-informative allocation often struggles to separate concepts, frequently producing hybrids (the left side of one object fused with the right side of another), whereas this behavior is rare for our method. The precomputed allocation reduces diversity, with objects repeatedly appearing in the same locations.
>
> **Q2: Additional Metrics**
> MultiDiffusion allows us to report an additional metric: the percentage of sampling timesteps with fairness-constraint violations. Note that we do not compute fairness criteria wrt the background player. Our method shows a consistent improvement over MultiDiffusion baseline highlighting the benefit of dynamic allocation that adapts to model preferences.
> Notice that averaging baseline never violates fairness constraints by definition. We do not compare with other baselines since computing a fairness constraint requires an allocation, which they do not provide.
>
> | Players | Coordination Strategy | Envy-free ↓ | Proportional ↓ | Equitable ↓ |
> | :---- | :---- | :---- | :---- | :---- |
> | 2 | Multidiffusion | 0.485 ± 0.233 | 0.485 ± 0.233 | 1.0 ± 0.00 |
> | 2 | Ours (without fairness) | 0.196 ± 0.118 | 0.306 ± 0.215 | 0.99 ± 0.001 |
> | 2 | Ours (with proportional constraints) | 0.015 ± 0.008 | 0.039 ± 0.05 | 0.99 ± 0.002 |
> | 3 | Multidiffusion | 0.845 ± 0.171 | 0.742 ± 0.217 | 1.0 ± 0.00 |
> | 3 | Ours (without fairness) | 0.356 ± 0.216 | 0.44 ± 0.229 | 0.99 ± 0.004 |
> | 3 | Ours (with proportional constraints) | 0.017 ± 0.010 | 0.042 ± 0.08 | 1.0 ± 0.00 |
>
> MultiDiffusion violates envy-freeness \~49% of the time with 2 players and \~85% with 3 players. This means at least one model is being frequently **neglected or dominated by other models**. Even without explicit fairness constraints, our efficient allocation reduces violations substantially. With proportional fairness constraints, envy-free violations drop to 1.5% (n=2) and 1.7% (n=3). This small percentage is caused by numerical approximations in the dual problem solution. Equitable constraints are violated most of the time. This is expected since proportionality does not imply equitability which has to be imposed separately. We hope the fairness violations table clarifies the advantage of dynamic allocation and quantifies how our method prevents model neglect.
>
> **Q3: Do allocations change…**
>
> As shown by Figure 2 from the main text with the bus-car example, the allocation changes mainly in the beginning of sampling and remains almost constant afterwards. This suggests that the fair division game can be solved only for the first k1 iterations with little effect on the performance. We validated this hypothesis with k1=15, denoted by Ours (k1=15, k2=50), cutting the time of fair division stage to half of what was previously reported, denoted by Ours (k1=50,k2=50). The results are reported in the table below.
>
> | Players | Coordination Strategy | GenEval (%images) | GenEval (%prompt) | CLIP (joint) | CLIP (min) | Reward (joint) | Reward (min) | VQA (joint) | VQA (min) |
> | :---- | :---- | :---- | :---- | :---- | :---- | :---- | :---- | :---- | :---- |
> | 2 | Ours (k1=15, k2=50) | 86.25% | 99.00% | 29.91 | 21.54 | 1.22 | \-0.39 | 0.958 | 0.922 |
> | 2 | Ours (k1=50, k2=50) | 88.50% | 99.00% | 30.02 | 21.53 | 1.23 | \-0.38 | 0.960 | 0.925 |
> | 3 | Ours (k1=15, k2=50) | 58.00% | 89.00% | 32.91 | 19.01 | 1.15 | \-0.83 | 0.906 | 0.812 |
> | 3 | Ours (k1=50, k2=50) | 59.50% | 92.00% | 33.21 | 19.09 | 1.22 | \-0.79 | 0.921 | 0.829 |
>
> **Q4: How costly is …**
> we combine this answer with the Q1-Q2 from reviewer vSpp. Please refer to it.

---

> > ### Author Rebuttal · Reviewer_sVXN · 2026-04-02
> >
> > Thank you for the detailed response.
> >
> > I believe that including the MultiDiffusion baseline really helps explain the advantages of your algorithm. I would urge you to include it in the main text of the final version, space permitting.
> >
> > Given that my concerns have been addressed I will be raising my score.

---

### Official Review · Reviewer_vJU4 · 2026-03-11

**Soundness:** 3
**Presentation:** 3
**Significance:** 2
**Originality:** 2
**Overall Recommendation:** 4
**Confidence:** 3

**Summary:**

This paper introduces Divide-and-Denoise, a training-free framework for coordinating multiple pre-trained diffusion models to generate multi-concept images. The core innovation is treating the composition of models as a fair division game. By alternating between (i) dividing latent space regions using game-theoretic fairness criteria and (ii) denoising those regions with assigned models, the method prevents common composition failures like missing objects or mismatched attributes. The authors provide two utility formulations—score-based and attention-based—making the method applicable across different architectures like Stable Diffusion and DiT.

**Compliance With Llm Reviewing Policy:**

Affirmed.

**Key Questions For Authors:**

See above.

**Limitations:**

Yes

**Strengths And Weaknesses:**

**Strength**

1. Applying game theory (fair division) to the challenge of model composition is a creative and theoretically grounded approach.

2. The method is fully compositional at inference time; models do not need to share weights or architectures.

3. The method significantly outperform baselines (Averaging, Composable Diffusion) on the GenEval benchmark, particularly in attribute binding and multi-object generation.

**Weakness**

1. While the framework is motivated by game theory, the paper appears to provide limited formal guarantees regarding convergence of the composite diffusion process, stability of the division procedure, optimality of the fairness criterion. The connection to formal game-theoretic equilibria could be strengthened.

2. Regional denoising can introduce boundary inconsistencies between regions assigned to different models. The paper could analyze whether this occurs and how it is mitigated.

3. The division step is central to the method but appears somewhat heuristic. How robust is the partitioning across timesteps? How sensitive is it to hyperparameters? Could partitions lead to artifacts or seams?

---

> ### Author Rebuttal · Authors · 2026-03-31
>
> **Q1: While the framework is motivated by game theory, the paper appears to provide limited formal guarantees…**
> As reviewer 1 mentioned, "The proposed game-theoretic view of composing multiple diffusion models has not been discussed in the literature before. It constitutes an interesting alternative to previous works that treated model composition through the lens of probability theory/energy functions." In other words, the aim of this paper is to rethink composition by defining an algorithmic procedure for combining models rather than defining a specific clean-time probability distribution that we would like to sample from. Elaborating on theoretical properties of the distribution of samples generated will require further work and potentially new mathematical machinery because of our modified sampling process at each step of generation.
>
> **Q2: Regional denoising can introduce boundary inconsistencies…**
>
> We rarely observe boundary inconsistencies when using our method. In fact, our method includes several mechanisms that mitigate boundary artifacts:
>
> 1. Notice how at every iteration each model takes as input the whole image, not just its assigned region. This ensures proposals adapt to what other models have generated.
> 2. Our allocations are soft, meaning that each latent coordinate is fractionally assigned to all models. Hence, boundary pixels are shared (see Figure 2 from the main text).
> 3. We explicitly encourage interactions between models by introducing a background player. As shown by the fish \+ anemone example in Figure 4, when utilities near a boundary are low, the background player takes over and enforces smooth transitions.
>
> That said, boundary inconsistencies can still occur even if rarely. In our experiments, they mainly arise when composing very different objects that were typically seen with different backgrounds during training (see the bird \+ hair dryer example in Table 10 in Appendix).
>
> **Q3: ... How robust is the partitioning across timesteps? How sensitive is it to hyperparameters? …**
>
> Since validation is expensive in our setup, we hand-picked hyperparameters $\\alpha$ and $\\beta$ used in the main text based on initial experiments. Below, we provide a more comprehensive discussion on hyperparameter effect and sensitivity. We will add it to the Appendix.
>
> Our method has two main hyperparameters: $\\beta\_t$, the regularization weight in the division problem, and $\\alpha\_t$, the regularization weight in the compositional update. We use a constant $\\beta\_t \= \\beta$ throughout generation, and we reparameterize $\\alpha\_t$ via a single scalar $\\alpha$ that controls the guidance strength. Specifically (**correcting a typo in the main text**),
>
> \\\[\\alpha\_t \= \\frac{\\sigma\_t}{\\alpha} \\left\\| \\nabla\_{x\_{t}} U\_{t}(x\_{t}, Q)\\right\\|\\\]
>
> We find it advantageous to match the guidance magnitude to the noise level, as overly strong guidance can drive intermediate latents off the training manifold. For a DDIM sampler with hyperparameter $\\eta$, this corresponds to setting $\\eta \= \\alpha$. Since DDIM typically performs best with $\\eta$ near zero, we focus on small $\\alpha$.
>
> Edge cases:
>
> $\\beta \\to \\infty$ yields a fixed allocation (with $\\alpha \= 0$ this recovers the averaging baseline).
>
> $\\beta \= 0$ yields an unregularized allocation that depends only on current utilities. Can cause numerical instability. Not robust to noise in utilities.
>
> $\\alpha \= 0$ disables guidance
>
> $\\alpha \\gg 0$ leads to off-manifold updates and artifacts.
>
> Observed trends:
>
> Guidance is not effective when the allocation is nearly uniform, which motivates using $\\beta$ in a lower range. At the same time, when utilities are noisy, stronger regularization is required to ensure that the regions are consistent across time, thus we use larger $\\beta$ for score-based utilities.
>
> Increasing $\\alpha$ from $0$ initially improves performance, but beyond a certain point performance drops due to artifacts from out-of-manifold updates.
>
> Overall, performance is fairly robust to small perturbations around the optimal ranges. Results for selected hyperparameter values are summarized in the table below (some numbers are omitted due to the character limit). Ablation uses our 2-object composition with Stable Diffusion setup. We note that different metrics might have different sensitivity to artifacts.
>
> | $\\alpha$ | $\\beta$ | GenEval %img | CLIP jnt | Reward jnt | VQA jnt |
> | :---- | :---- | :---- | :---- | :---- | :---- |
> | 0.015 | 0.001 | 88.50% | 30.02 | 1.23 | 0.960 |
> | 0.01 | 0.001 | 87.50% | 29.77 | 1.15 | 0.954 |
> | 0.02 | 0.001 | 88.50% | 30.02 | 1.25 | 0.963 |
> | 0.05 | 0.001 | 88.25% | 30.10 | 1.18 | 0.956 |
> | 0.1 | 0.001 | 78.75% | 29.95 | 0.86 | 0.927 |
> | 0.015 | 0.0005 | 89.00% | 30.01 | 1.24 | 0.960 |
> | 0.015 | 0.002 | 88.00% | 29.96 | 1.23 | 0.965 |
> | 0.015 | 0.01 | 84.00% | 29.82 | 1.13 | 0.960 |

---

> > ### Author Rebuttal · Reviewer_vJU4 · 2026-04-03
> >
> > The rebuttal addressed most of concerns.

---

### Official Review · Reviewer_vSpp · 2026-03-12

**Soundness:** 3
**Presentation:** 3
**Significance:** 3
**Originality:** 4
**Overall Recommendation:** 5
**Confidence:** 3

**Summary:**

This paper tackles the problem of composing multiple pre-trained diffusion models without ground-truth partitions. The authors propose Divide-and-Denoise, a game-theoretic framework that couples a division process with a composite denoising process. Key contributions include an inference-time algorithm for coordinating models, formulations for model utilities, and empirical validation across Stable Diffusion and DiT models. Experimental results demonstrate that the method outperforms selected baselines in multi-concept generation, attribute binding, and conflicting concept composition.

**Compliance With Llm Reviewing Policy:**

Affirmed.

**Final Justification:**

The authors addressed most of the concerns raised in the initial review, and properly answered follow-up questions. The additional results and the authors' clarifications strengthen the submission. The strengths of this work now outweigh its weaknesses and limitations. All factors considered, I raise my score from 4 to 5 to recommend acceptance.

**Key Questions For Authors:**

Please see Strengths and Weaknesses.

**Limitations:**

Limitations such as computational overheads should be explicitly discussed.

**Strengths And Weaknesses:**

## Strengths

1. The integration of fair division and composable diffusion is conceptually novel and interesting, and the authors empirically demonstrate the strengths of this combination.
2. The proposed method appears technically sound, as it is grounded in fair division theory and several theoretical justifications are provided.
3. Experimental results show strong performance across three aspects: multi-concept generation, attribute binding, and conflicting concept composition.
4. Useful ablations are performed across guidance steps, fairness constraints, and utilities.


## Weaknesses

1. The method introduces noticeable computational overhead (\~4$\times$ in runtime) compared to composable diffusion and other naive baselines, as reported in Table 5 in the appendix.
    - Are there any practical techniques to mitigate this issue?
2. The time and space complexity of the proposed method are not provided, and a detailed discussion of scalability is missing.
    - A breakdown of the runtime of different modules/stages/processes in an entire inference run (from noise to clean sample) would be useful.
    - Could the authors provide an ablation (even at small scale) for composing more than 3 models and report the corresponding runtime and memory usage to demonstrate how the method scales with more models?
3. Hyperparameter sensitivity and selection are not adequately discussed or ablated.
4. The evaluation does not fully validate the fairness-centric claims. The paper argues that fairness prevents neglect and domination, but the empirical section does not report direct fairness statistics such as envy violations, proportionality satisfaction, etc.
5. Variances are not reported for any of the quantitative results.
6. Limitations of the method are not explicitly discussed.
7. Reproducibility is limited: no code implementation or anonymous repository is provided.

---

> ### Author Rebuttal · Authors · 2026-03-31
>
> **Q1: Time and Space Complexity. A breakdown of the runtime of different modules/stages/processes …**
>
> We first report a breakdown of wall-clock time and CPU memory usage of our method with a batch size 1 as we scale the number of models from 2 to 4. Our method contains three stages as described by Algorithm 1 in the main text: proposals (where we aggregate each model’s denoising kernel and utilities), division (where we solve a fair division game given by a convex optimization problem), and denoising (where we compute the composite denoising kernel). We report what fraction of total generation time is consumed by each stage. We also report what fraction of total CPU memory used during generation is consumed by each stage.
>
> | Players | Proposal (%) | Division (%) | Denoising (%) | Proposal Mem (%) | Division Mem (%) | Denoising Mem (%) |
> | :---: | :---: | :---: | :---: | :---: | :---: | :---: |
> | 2 | 0.404±0.06 | 0.118±0.13 | 0.462±0.07 | 0.170±0.065 | 0.820±0.069 | 0.013±0.011 |
> | 3 | 0.298±0.09 | 0.300±0.23 | 0.382±0.12 | 0.075±0.064 | 0.918±0.070 | 0.007±0.009 |
> | 4 | 0.174±0.10 | 0.569±0.25 | 0.244±0.14 | 0.031±0.042 | 0.964±0.047 | 0.004±0.006 |
>
> The Division stage is the primary bottleneck as the number of players grows. It increases from 12% to 57% of total time between 2 and 4 players, with a large variance in these rates from generation to generation. It also dominates CPU memory, reaching 96% at 4 players. The Denoising stage is the most time consuming at 2 players but its relative share decreases with more players.
>
> We note that the Division stage consumes the majority of the CPU memory because it is the only instance where we perform operations on the CPU instead of the GPU. Specifically, the Proposal stage runs a forward pass for each diffusion model and saves model utilities, all of which happens on the GPU. The Denoising stage does not consume any significantly additional memory. Finally, the Division stage uses an off-the-shelf convex optimization solver to solve the fair division game, all of which happens on the CPU.
>
> **Q2: … Are there any practical techniques to mitigate this issue? …**
>
> From the first table, the main bottlenecks are the division step and the denoising step (due to the guidance term). We note that image structure is fixed early in generation, making both guidance and division less effective over time. Thus, to speed up with low effect on performance, we can limit division and guidance to the first k1 and k2 steps, respectively. We conduct an ablation on k1 and k2 and report the average time per stage. Division time drops significantly when k1=15. See in our response to reviewer sVXN that metrics change only slightly.
>
> | Players | Coordination Strategy | proposal (ms) | division (ms) | denoising (ms) | total (ms) |
> | :---- | :---- | :---- | :---- | :---- | :---- |
> | 2 | Baseline | \-- | \-- | \-- | 78.06 ± 15.44 |
> | 2 | Ours (k1=50, k2=50) | 68.18 ± 0.93 | 29.36 ± 67.54 | 77.88 ± 0.218 | 175.42 ± 67.55 |
> | 2 | Ours (k1=15, k2=50) | 68.06 ± 0.793 | 16.38 ± 20.75 | 77.82 ± 0.24 | 162.26 ± 20.77 |
> | 2 | Ours (k1=50, k2=15) | 64.62 ± 0.883 | 104.09 ± 137.21 | 23.41 ± 0.198 | 192.12 ± 137.21 |
> | 2 | Ours (k1=15, k2=15) | 65.12 ± 0.738 | 17.36 ± 21.83 | 23.72 ± 0.780 | 106.20 ± 21.86 |
> | 3 | Baseline | \-- | \-- | \-- | 85.03 ± 21.81 |
> | 3 | Ours (k1=50, k2=50) | 79.67 ± 0.95 | 131.72 ± 175.9 | 101.97 ± 0.258 | 313.36 ± 175.90 |
> | 3 | Ours (k1=15, k2=50) | 80.9 ± 0.851 | 72.41 ± 70.11 | 102.56 ± 0.278 | 255.87 ± 70.12 |
> | 3 | Ours (k1=50, k2=15) | 76.05 ± 0.944 | 277.25 ± 270.08 | 30.88 ± 0.176 | 384.18 ± 270.08 |
> | 3 | Ours (k1=15, k2=15) | 79.54 ± 0.962 | 71.46 ± 68.37 | 30.66 ± 0.178 | 181.66 ± 68.38 |
> | 4 | Baseline | \-- | \-- | \-- | 92.58 ± 24.26 |
> | 4 | Ours (k1=50, k2=50) | 90.82 ± 0.922 | 524.54 ± 450.32 | 126.48 ± 0.263 | 741.84 ± 450.32 |
> | 4 | Ours (k1=15, k2=50) | 90.6 ± 1.02 | 255.21 ± 143.48 | 126.56 ± 0.251 | 472.37 ± 143.48 |
> | 4 | Ours (k1=50, k2=15) | 86.03 ± 1.04 | 746.9 ± 500.4 | 38.1 ± 0.179 | 871.03 ± 500.40 |
> | 4 | Ours (k1=15, k2=15) | 86.18 ± 0.816 | 255.06 ± 144.752 | 38.13 ± 0.171 | 379.37 ± 144.75 |
>
> The cost grows greatly with $n$, mainly due to the division step. This step could be accelerated by using a faster solver, relaxing the tolerance, or fixing the number of solver iterations.
>
> **Q3: Hyperparameter sensitivity …**
> We combine this answer with Q3 from reviewer vJU4. Please refer to it.
>
> **Q4: ... the fairness-centric claims ...**
> We combine this answer with the Q2 from reviewer sVXN. Please refer to it.
>
> **Q5: Variances are not reported …**
> We will add variances to all metrics in the revised text.
>
> **Q6: Limitations …**
> Our method cannot be used for compositional tasks where blending of concepts is expected. If asked to combine “a dog” and “a watercolor painting”, our method will not generate a dog painted with the watercolor but two separate objects.
>
> **Q7: Reproducibility is limited …**
> We will upload code upon acceptance

---

> > ### Author Rebuttal · Reviewer_vSpp · 2026-04-01
> >
> > Thanks for the detailed response. My concerns are mostly addressed, with only one minor follow-up question.
> > - Could you also disclose the task performance when applying the technique proposed in your response to Q2? This should verify the "speed up with low effect on performance" claim.
> >
> > I will raise my score if that claim is supported by quantitative results.

---

> > > ### Author Response · Authors · 2026-04-07
> > >
> > > As requested, we provide performance metrics for the n-object composition task described in Section 4.4 Stable Diffusion. We keep all hyperparameters fixed while decreasing the number of division steps k1 or/and guidance steps k2. Earlier we reported that wall-clock time of Divide-and-Denoise may be improved considerably by selecting k1=15, k2=15. The performance metrics for each (k1,k2) are presented in the table below. We also report the metrics for the strongest baseline in this table for comparison.
> > >
> > > | Players | Coordination Strategy | GenEval %images | GenEval %prompt | CLIP joint | CLIP min | Reward joint | Reward min | VQA joint | VQA min |
> > > |---|---|---|---|---|---|---|---|---|---|
> > > | 2 | Multi-Concept Diffusion | 53.75% | 86.00% | 27.05 | 18.77 | 0.280 | -1.150 | 0.753 | 0.683 |
> > > | 2 | Multidiffusion | 58.00% | 93.00% | 27.65 | 19.59 | 0.340 | -0.990 | 0.816 | 0.738 |
> > > | 2 | Ours (k1=50, k2=50) | **88.50%** | **99.00%** | **30.02** | 21.53 | **1.230** | **-0.380** | **0.960** | **0.925** |
> > > | 2 | Ours (k1=15, k2=50) | 86.25% | 99.00% | 29.91 | 21.54 | 1.220 | -0.390 | 0.958 | 0.922 |
> > > | 2 | Ours (k1=50, k2=15) | 85.25% | 99.00% | 29.79 | 21.38 | 1.178 | -0.417 | 0.953 | 0.910 |
> > > | 2 | Ours (k1=15, k2=15) | 83.25% | 99.00% | 29.77 | 21.36 | 1.148 | -0.435 | 0.953 | 0.910 |
> > > | 3 | Multi-Concept Diffusion | 14.75% | 43.00% | 28.45 | 15.15 | -0.140 | -1.820 | 0.532 | 0.384 |
> > > | 3 | Multidiffusion | 14.00% | 37.00% | 28.04 | 16.02 | -0.477 | -1.789 | 0.536 | 0.374 |
> > > | 3 | Ours (k1=50, k2=50) | **59.50%** | **92.00%** | **33.21** | **19.09** | **1.220** | **-0.790** | **0.921** | **0.829** |
> > > | 3 | Ours (k1=15, k2=50) | 58.00% | 89.00% | 32.91 | 19.01 | 1.150 | -0.830 | 0.906 | 0.812 |
> > > | 3 | Ours (k1=50, k2=15) | 50.00% | 88.00% | 32.53 | 18.60 | 1.072 | -0.901 | 0.864 | 0.761 |
> > > | 3 | Ours (k1=15, k2=15) | 48.50% | 86.00% | 32.34 | 18.47 | 0.999 | -0.860 | 0.846 | 0.737 |
> > >
> > > In line with our observation that allocation is mainly determined early in the generation, we only notice slight decrease in metrics with k1=15. We observe that stopping guidance after k2=15 steps has minor effect for n=2, but a larger effect when number of players increase. This trend likely arises because the importance of guidance increases in the presence of conflicts between concepts, which is common in our setup of sampling n random objects and becomes more frequent as n grows. We expect to see even less effect on natural tasks. Crucially, we still reliably outperform the baselines with persistent and large margins.

---

### Decision · Program_Chairs · 2026-04-30

**Decision:**

Accept (spotlight)

**Comment:**

The paper proposes a new sampling method for coordinating multiple pre-trained diffusion models. The reviewers unanimously support acceptance, highlighting its novelty, solid theoretical grounding, and strong experimental results. AC agrees that the paper introduces a novel and well-founded idea that is of clear value to the community, and therefore recommends acceptance.